# Two RhoGEF isoforms with distinct localisation control furrow position during asymmetric cell division

Emilie Montembault [1,3,5], Irène Deduyer [1,3,5], Marie-Charlotte Claverie [1,3,5], Lou Bouit[1,4], Nicolas J. Tourasse [2], Denis Dupuy[2], Derek McCusker [1,3] & Anne Royou [1,3] ✉

Cytokinesis partitions cellular content between daughter cells. It relies on the formation of an acto-myosin contractile ring, whose constriction induces the ingression of the cleavage furrow between the segregated chromatids. Rho1 GTPase and its RhoGEF (Pbl) are essential for this process. However, how Rho1 is regulated to sustain furrow ingression while maintaining correct furrow position remains poorly defined. Here, we show that during asymmetric division of *Drosophila* neuroblasts, Rho1 is controlled by two Pbl isoforms with distinct localisation. Spindle midzone- and furrow-enriched Pbl-A focuses Rho1 at the furrow to sustain efficient ingression, while Pbl-B pan-plasma membrane localization promotes the broadening of Rho1 activity and the subsequent enrichment of myosin on the entire cortex. This enlarged zone of Rho1 activity is critical to adjust furrow position, thereby preserving correct daughter cell size asymmetry. Our work highlights how the use of isoforms with distinct localisation makes an essential process more robust.

Cell cleavage, also called cytokinesis, is a fundamental process that occurs subsequent to sister chromatid segregation. In animal cells, cytokinesis relies on the assembly of a contractile ring composed of filamentous actin and bipolar filaments of non-muscle myosin II (referred to as myosin) anchored at the equatorial plasma membrane. Myosin activity generates the force necessary to drive the ingression of the cleavage furrow. The topological resolution of the two resulting cells is achieved by abscission of the membrane[1,2]. The activity of the small GTPase RhoA (Rho1 in *Drosophila*) is essential for this process as it promotes actin nucleation and myosin activation[3–6]. Rho1 is a small GTPase of the Ras super-family that undergoes nucleotide-induced conformational changes as it toggles between inactive GDP- and active GTP-bound states. Rho1 localises to the plasma membrane through prenylation at its carboxy-terminus and its lipid interaction is obligatory for function[7]. The zone of active Rho1 at the plasma membrane

determines the position of contractile ring assembly and subsequent cleavage furrow ingression[8,9]. Rho1 activation is catalysed by a guanine nucleotide exchange factor (GEF) called Ect2 (Epithelial cell transforming 2) in mammals[10–13] and Pebble (Pbl) in *Drosophila*[5,14,15]. Hence, the mechanism controlling Ect2/Pbl localisation and activation is critical to determine the zone of Rho1 activation and thus the position of the resulting furrow. Ect2/Pbl contains BRCT (BRCA1 C-terminal) domains in its N-terminus and a catalytic DH (Dbl homology) domain and juxtaposed membrane-associated PH (Plekstrin homology) domain in its C-terminus. The canonical mechanism by which Ect2/Pbl concentrates at the equatorial zone to activate Rho1 is through its interaction with mgcRacGAP/Cyk4 (RacGAP50C or tumbleweed in *Drosophila*) via the BRCTs[16–18]. RacGAP50C forms part of the conserved centralspindlin complex with the kinesin 6 MKLP1/Zen4 (Pavarotti in *Drosophila*). This complex promotes the bundling of anti-parallel

[1]CNRS, UMR5095, University of Bordeaux, Institut Européen de Chimie et Biologie, 2 rue Robert Escarpit, 33607 Pessac, France. [2]University of Bordeaux, INSERM, U1212, Institut Européen de Chimie et Biologie, 2 rue Robert Escarpit, 33607 Pessac, France. [3]Present address: CNRS, UMR5095, University of Bordeaux, Institut de Biologie et Génétique Cellulaire, 1 rue Camille Saint-Saëns, 33077 Bordeaux, France. [4]Present address: CNRS, UMR5297, University of Bordeaux, 146 Rue Léo Saignat, 33076 Bordeaux, France. [5]These authors contributed equally: Emilie Montembault, Irène Deduyer, Marie-Charlotte Claverie. ✉e-mail: anne.royou@u-bordeaux.fr

microtubules at their plus-end to form the midzone of the central spindle that assembles between the two segregated pools of chromatids[11,14,16,17,19,20]. In vertebrates, the phosphorylation of mgcRacGAP by Polo-like Kinase 1 provides docking sites for Ect2's BRCT domains, thereby relieving Ect2 auto-inhibition and promoting its accumulation at the spindle midzone and equatorial membrane[21–24].

While the molecular pathway that specifies the zone of Rho1 activation to initiate contractile ring assembly is well defined, less is known about the mechanisms that maintain the contractile ring position while sustaining efficient constriction. The study of these mechanisms is hindered by the fact that severe perturbation of either Rho1 signalling or the kinetics of contractile ring components during division results in failure to initiate furrowing or furrow regression. However, the mechanism that controls Rho1 activity to sustain furrowing at the right position is critical to preserve the size of the resulting daughter cells. This is particularly important during asymmetric stem cell division where cell size determines cell fate, as observed in the *C. elegans* Q neuroblast lineage, where the smaller daughter cells undergoes apoptosis[25].

The *Drosophila* larval neuroblast is a powerful model for studying the mechanisms that spatio-temporally control the asymmetric position of the furrow. The division of this large stem cell produces two cells that differ in size and fate: the larger cell retains the neuroblast "stemness" and continues dividing frequently, while the smaller ganglion mother cell (GMC) undergoes one round of division before differentiating into neuronal lineages[26]. Previous studies have shown that the asymmetric position of the furrow is specified by two pathways, a spindle-independent pathway that requires the polarity complex Pins (composed of Partner of Inscuteable (Pins); Gαi; Discs Large) to bias Myosin activity towards the basal cortex and a microtubules-dependent pathway that regulates RacGAP50C localisation to the equatorial cortex[27–35].

In addition to these pathways, our team recently identified novel Pbl-dependent myosin dynamics during neuroblast division. At mid-ring closure, a pool of myosin undergoes outward flow (efflux) from the contractile ring, invading the entire cortex (hereafter referred to as the "polar" cortex or the "poles", as opposed to the furrow) until the end of furrow ingression[36]. This pool of active myosin at the poles becomes critical if chromatids trail at the cleavage site during furrow ingression. Under these conditions, cell elongation ensues from the reorganisation of polar myosin into broad lateral rings that partially constrict. This "adaptive" cell elongation facilitates the clearance of trailing chromatids from the cleavage site. Attenuation of Pbl function impairs myosin polar cortex dynamics during division. Consequently, cells cannot undergo adaptive elongation in the presence of trailing chromatids, with dire consequences for tissue homeostasis[36,37]. While these Pbl-dependent changes in the mechanical properties of the polar cortex are important for the clearance of trailing chromatids, little is known about their role in maintaining the fidelity of asymmetric neuroblast division under normal conditions.

In this study, we investigate the consequences of perturbed myosin polar cortex enrichment due to reduced Pbl activity during asymmetric cell division by live imaging of neuroblasts. We found that in the absence of myosin polar cortex activity associated with the *pbl^{MS}* mutation, defects in daughter cell size asymmetry arise during division. We identified *pbl^{MS}* as a non-sense substitution in an alternatively spliced exon. The mutation prevents the production of Pbl-B, one of two major Pbl isoforms that we identify. However, the *pbl^{MS}* mutation does not affect Pbl-A, the sole Pbl isoform reported in the literature thus far. We found that unlike Pbl-A, Pbl-B is not enriched at the central spindle midzone and furrow. Additionally, it is sequestered in the nucleus prior to Pbl-A. Genomic rescue experiments revealed that the lack of a functional Pbl-B recapitulates *pbl^{MS}* cytokinesis phenotypes including the lack of myosin polar cortex enrichment and the formation of abnormally small GMC.

In contrast, the lack of Pbl-A expression induced excessive myosin enrichment to the polar cortex associated with a transient but dramatic widening of the furrow that mildly affected its rate of ingression. These phenotypes are similar to those observed after depletion of RacGAP50C. We propose that two Pbl isoforms displaying distinct localisation patterns modulate Rho1 activity concurrently to adjust furrow position during ingression and maintain cell size asymmetry.

## Results

### Myosin does not enrich the nascent daughter cell cortex during furrow ingression in *pbl^{MS}* mutant neuroblasts

To determine the importance of myosin and, hence, Rho1 activity at the poles during asymmetric cell division, we examined myosin dynamics during *Drosophila* larval neuroblast cytokinesis in wild type (WT) and a homozygote mutant for an hypomorph allele of *pbl* called *pbl^{MS}*. In WT neuroblasts, myosin adopted a uniform cortical localisation at anaphase onset (initiation of sister chromatid separation) (Fig. 1a, Supplementary Movie 1). Subsequently, myosin depleted the apical then the basal pole and enriched the baso-lateral cortex to initiate furrowing as previously described (Fig. 1a and Supplementary Fig. 1)[28,30,32,36]. Shortly after the onset of furrow ingression (defined as the onset of inward membrane curvature), a pool of myosin underwent outward flow from the ring and enriched the entire cortex of both nascent daughter cells (Fig. 1a)[36]. Myosin concentration at the poles was maximal at approximately 80% of furrow ingression (Fig. 1a–e). Myosin dissociated entirely from the cortex concomitantly with nuclear envelope reassembly (as illustrated by the depletion of myosin from the cortex and its exclusion from the nascent nucleus) (Fig. 1a)[36]. In contrast, no myosin signal was detected at the polar cortex at any time points during cytokinesis in the *pbl^{MS}* homozygote mutant, reminiscent of myosin dynamics observed in the *pbl^{5}/pbl^{MS}* transheterozygote mutant (Fig. 1a–e, Supplementary Movie 1)[36]. After apical and basal depletion, myosin rapidly compressed into a tight ring at the baso-lateral position, where it remained focused throughout closure (Fig. 1a–e and Supplementary Fig. 1).

### The *pbl^{MS}* mutant exhibits defects in daughter cell size asymmetry

Next, we determined if the absence of active myosin at the polar cortex affected the efficiency of contractile ring constriction. To do so, we measured the diameter of the contractile ring over time from anaphase onset. We considered that 100% constriction occurred when the ring reached the midbody stage (the midbody is defined as the remnant of the contractile ring and midzone components at completion of ring closure)(Fig. 1f). No difference in the rate of ring constriction was observed between WT and *pbl^{MS}*, suggesting that the outward flow of myosin from the contractile ring and transient activity at the polar cortex was not important for efficient furrow invagination.

Next, we examined if the position of the furrow was maintained during ring constriction in *pbl^{MS}*. To do so we first measured the area of the resulting GMC at the end of ring closure. Remarkably, a significant proportion of *pbl^{MS}* cell divisions led to the production of abnormally small GMCs, suggesting a defect in furrow position during ingression (Fig. 1g, h). To determine if the abnormally small GMC size was due to furrow mis-positioning at an early stage during cleavage, we measured furrow position at the onset of ingression. We found that, as previously reported, the furrow was positioned towards the basal pole in WT cells (Fig. 1i)[28–30,32]. No difference in initial furrow position was observed in *pbl^{MS}* compared to WT, indicating that the small GMC phenotype in *pbl^{MS}* was not due to early mis-positioning of the furrow (Fig. 1i). We noticed some variation in the initial basal position of the furrow in both WT and *pbl^{MS}* cells. Since not all *pbl^{MS}* divisions produce abnormally small GMCs, we reasoned that the initial basal position of the furrow determined the size of the resulting GMC in *pbl^{MS}*. Thus, we plotted the position of

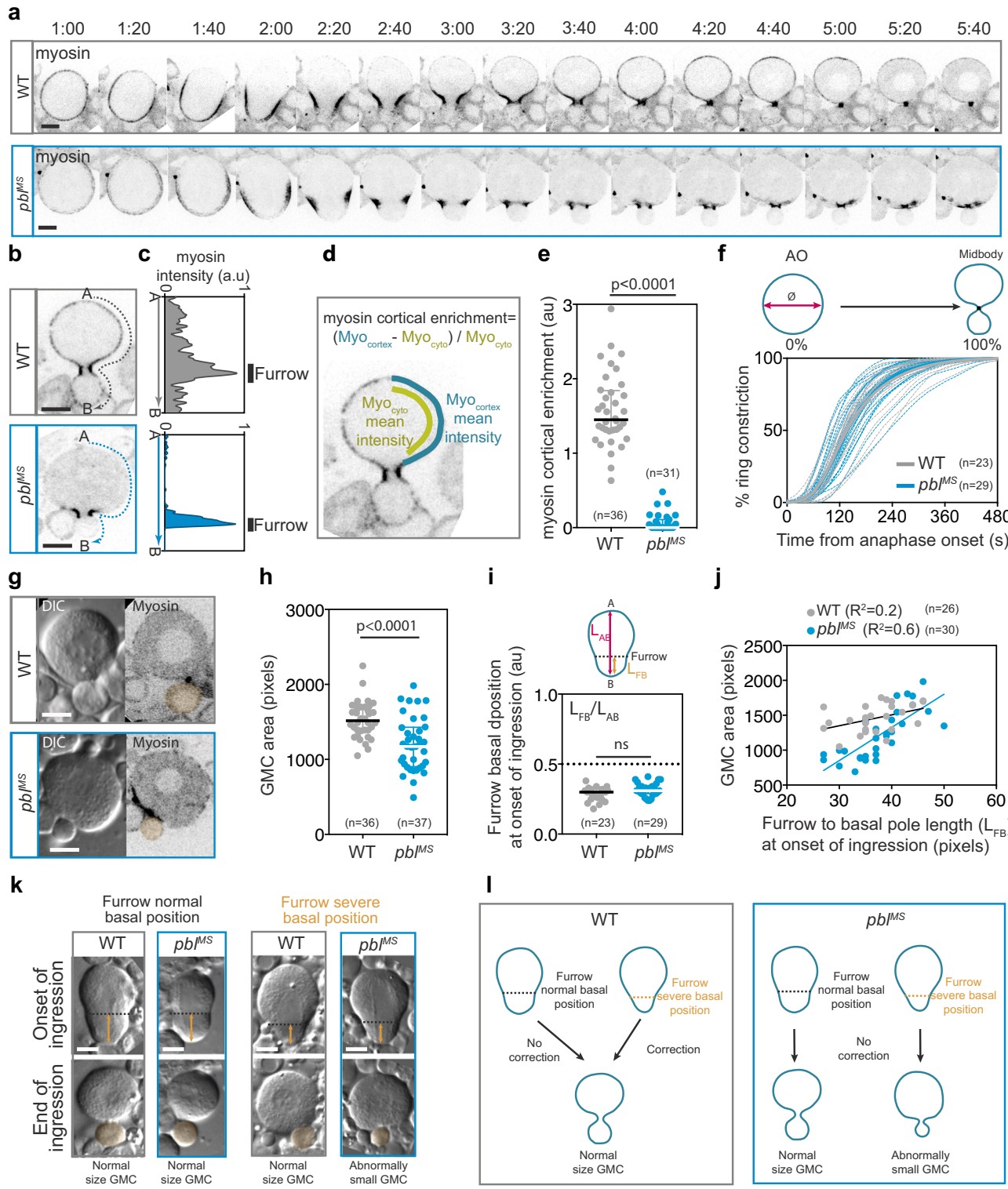

the furrow at ingression onset (as the distance from the furrow to the basal pole) with the area of the resulting GMC. Strikingly, in contrast to WT cells, a strong correlation was found between the initial position of the furrow and the size of the resulting GMC in $pbl^{MS}$ (Fig. 1j, k). These results are consistent with a model in which, in WT, the variation in the initial position of the furrow is sensed so that severe basal positions are corrected during ring constriction, thereby preserving the size of the resulting GMC. This correction mechanism does not appear to exist in the $pbl^{MS}$, hence furrows that are initially in a severe basal position produce abnormally small GMCs (Fig. 1l).

**The $pbl^{MS}$ mutation corresponds to a non-sense substitution in an alternatively spliced exon affecting the expression of Pbl-B, one of the two major Pbl isoforms expressed in *Drosophila***

The $pbl^{MS}$ allele was identified through an EMS screen for male sterility due to defects in spermatocyte division[38]. Since the mutation responsible for the $pbl^{MS}$ phenotype has not yet been characterised, we sequenced all the exons of the $pbl$ gene (CG8114) from $pbl^{MS}$ male adult genomic DNA extracts and compared these with the $pbl$ sequence of the wild type strain we had in the lab. The Flybase database annotation reports that the $pbl$ gene is encoded by 15 exons spread over 15 kb on

**Fig. 1 | Defects in cortical myosin dynamics and daughter cell size asymmetry during cytokinesis in *pbl^MS* mutant neuroblasts. a** Time-lapse images of wild type (WT) and *pbl^MS* mutant third instar larvae neuroblasts expressing Sqh::GFP (myosin). **b** Sagittal images of WT and *pbl^MS* cells at a similar timepoint during cytokinesis and (**c**) the corresponding linescan of myosin average intensity at the cortex from apical (A) to basal (B) poles (number of cells, $n = 36$ and $n = 31$ for WT and *pbl^MS*, respectively). **d** Scheme showing the method used to quantify the enrichment of myosin at the polar cortex and (**e**) the corresponding scatter dot plot. **f** Percentage of ring constriction over time from anaphase onset (AO) to midbody stage (100% constriction). Dashed and solid lines correspond to individual cells and the average, respectively. **g** DIC and the corresponding GFP (myosin) images of neuroblasts at the end of furrow ingression. GMC area are highlighted in orange in the GFP channel. **h** Distribution of the GMC area. **i** Distribution of the furrow basal position at onset of ingression calculated as illustrated in the scheme above the graph. **j** Graph showing the correlation between the furrow basal position at onset of ingression and the size of the resulting GMC at the end of ingression. Pearson r

correlation coefficient was used to calculate $R^2$. A significant positive correlation is found in *pbl^MS* mutants ($P < 0.0001$, two-tailed). **k** DIC images of cells at onset (top) and end (bottom) of furrow ingression. The left and right panels are representative examples of cells with normal or severe basal furrow position, respectively at onset of ingression. The dashed black line indicates the position of the furrow and the orange double arrows indicate the distance between furrow and basal pole. Number of cell, $n = 26$ and $n = 30$ for WT and *pbl^MS*, respectively. **l** Model for the abnormally small GMC phenotype resulting from *pbl^MS* mutant neuroblast divisions. In WT and *pbl^MS* mutant cells, the furrow is positioned towards the basal pole at the onset of ingression. Some furrows are positioned more basally than others. In WT but not *pbl^MS* mutant cells, this severe basal position is corrected, thereby preserving GMC size. Scale bars, 5 μm. Time, min:sec. Time, 0:00 corresponds to anaphase onset. For all scatter dot plots, bars correspond to median ± interquartile range. n number of cells. A Mann–Whitney two-tailed test was used to calculate P values.

---

chromosome 3 L (cytologic position 66A18-66A9 and genomic position 7896109-7911954). We identified a cytosine to thymidine substitution in exon 8 at position 7901087 on chromosome 3 L in *pbl^MS* DNA extracts from three independent experiments. This substitution changes the glutamine codon CAG into the stop codon TAG (Fig. 2a).

The Flybase website reports that exon 8 is alternatively spliced and is present in two of the six predicted *pbl* protein isoforms, Pbl-B and D isoforms (uniprot: Q8IQ97 for Pbl-B) (Fig. 2b). Hence, *pbl^MS* mutant flies do not express functional Pbl-B and D. To determine the overall level of the six predicted *pbl* isoform mRNAs we retrieved the raw read sequences from 1959 RNA-seq experiments publicly available in the NCBI SRA database [https://www.ncbi.nlm.nih.gov/sra] and mapped them to the *Drosophila* genome using the software HISAT2 in order to quantify relative usage of exon junctions of the *pbl* mRNA gene[39,40]. This large-scale quantitative analysis revealed that the major *pbl* mRNA isoforms detected overall and in various tissues are *pbl-RA* and *RB* (Fig. 2b and Supplementary Table 1). In addition, they are present at near equal amounts (Fig. 2c and Supplementary Table 1). Next, we determined the relative mRNA levels of these two major *pbl* isoforms in the larval CNS and adult testes. To do so we performed a semi-quantitative RT-PCR on mRNA samples extracted from third instar larval brains using primers specific to *pbl-RA* and *RB*. Our results revealed that *pbl-RB* is expressed at similar levels as *pbl-RA* in larval CNS and is slightly more abundant than *pbl-RA* in adult testes, consistent with our results from the Pbl exon junction usage analysis (Fig. 2c, Supplementary Table 1). In agreement, northern blots with *Drosophila* embryonic and adult mRNA extracts using *pbl* cDNA as a probe detected two major *pbl* transcripts of similar levels, a lower transcript corresponding to *pbl-RA* and a higher transcript consistent with the size of *pbl-RB*[41].

Next, we determined the amount of Pbl-A and B proteins in the larval brain. We were not able to detect both endogenous Pbl-A and B proteins by Western blot using different anti-Pbl antibodies. Therefore, to estimate the level of Pbl-A and B protein produced from alternative splicing we used transgenic flies containing the *pbl* genomic sequence tagged with GFP (Fig. 2d). Levels of GFP::Pbl-A and -B proteins were assessed by Western blot using anti-GFP antibodies on third instar larval brain extracts (Fig. 2e). Two bands corresponding to the expected size for GFP::Pbl-A and GFP::Pbl-B were detected only in the transgenic strain extract, indicating that the two GFP::Pbl proteins are produced at similar levels via alternative splicing. It is thus likely that the two endogenous Pbl isoforms are expressed at equal levels in this tissue. Collectively, these results indicate that Pbl-A and -B are the two major Pbl splicing variants expressed overall. In addition, they are present at similar levels in the central nervous system. We conclude that, since *pbl^MS* cells do not express Pbl-B, it is likely that Pbl-B but not Pbl-A drives myosin polar cortex enrichment during cytokinesis and that Pbl-B is important to

maintain the correct size of daughter cells during asymmetric cell division.

To examine the conservation of exon 8, we computed a multiple sequence alignment of 948 Pbl protein orthologs available in the OrhoDB database[42]. As expected, the alignment identified high conservation of the whole Pbl sequence amongst Eukaryotes with the exception of the region spanning exon 8. The sequence including exon 8 is found in a subset of 233 sequences belonging almost exclusively to the insecta class (Supplementary Fig. 2). Furthermore, a short region within the exon 8 sequence is shared between *Drosophilidae* and *Culicidae* (Supplementary Fig. 3). Finally, amongst *Drosophilidae*, exon 8 shows higher diversity than the rest of the Pbl sequence and is present in 36 out of 38 *Drosophilidae* species (Supplementary Fig. 4).

Next, we examined the differences in secondary structure predicted for Pbl-A and Pbl-B. Alphafold predicted a third BRCT domain (named BRCT0) in addition to the conserved N-terminal BRCT1 and 2 in Pbl-A (Fig. 2f) (https://alphafold.ebi.ac.uk/entry/Q9U7D8)[43]. This is consistent with a recent phylogeny study on the functional evolution of BRCT domains, which identified a third BRCT in the N-terminus of Ect2[44]. The structural analysis of the N-terminus of Ect2 confirmed the presence of three apposed BRCT domains[45]. While there is evidence that two BRCT domains are involved in Pbl and human Ect2 binding with RacGAP50C/MgcRacGAP, less is known about the role of the third BRCT (BRCT2) in this interaction[5,22–24,45]. We found that this sequence is inserted in the middle of the 76 amino acid region that is predicted to form the BRCT2 domain in Pbl-B (Fig. 2f). Alphafold modelling of Pbl-B predicted that the sequence encoded within exon 8 is unstructured but does not seem to hinder the folding of the BRCT2 (https://alphafold.ebi.ac.uk/entry/Q8IQ97) (Fig. 2f)[43]. Since the folding of the three BRCTs are predicted to be preserved in Pbl-B, we anticipated that Pbl-A and B will exhibit similar dynamics during neuroblast division.

## Pbl-B, unlike Pbl-A and RacGAP50C, is not enriched at the midzone and cleavage furrow

Next, we investigated Pbl-B and Pbl-A sub-cellular localisation during cytokinesis. To do so, we produced transgenic fly strains expressing GFP::Pbl-A or B under the control of the endogenous *pbl* promoter. We monitored GFP::Pbl-A or GFP::Pbl-B signals during cytokinesis in live neuroblasts and compared their localisation with VenusFP::RacGAP50C, the centralspindlin component known to interact with Pbl-A (Fig. 3a, Supplementary Movie 2)[19]. Recent studies have reported that, in addition to its localisation to the midzone, the centralspindlin component RacGAP50C's partner Pavarotti accumulates at the furrowing site via its association with peripheral astral microtubules[31]. In agreement with Pavarotti localisation, we detected VenusFP::RacGAP50C at the midzone and at the furrow throughout ingression. VenusFP::RacGAP50C strongly labelled the midbody upon completion of ring closure. Previous work reported that Pbl-A

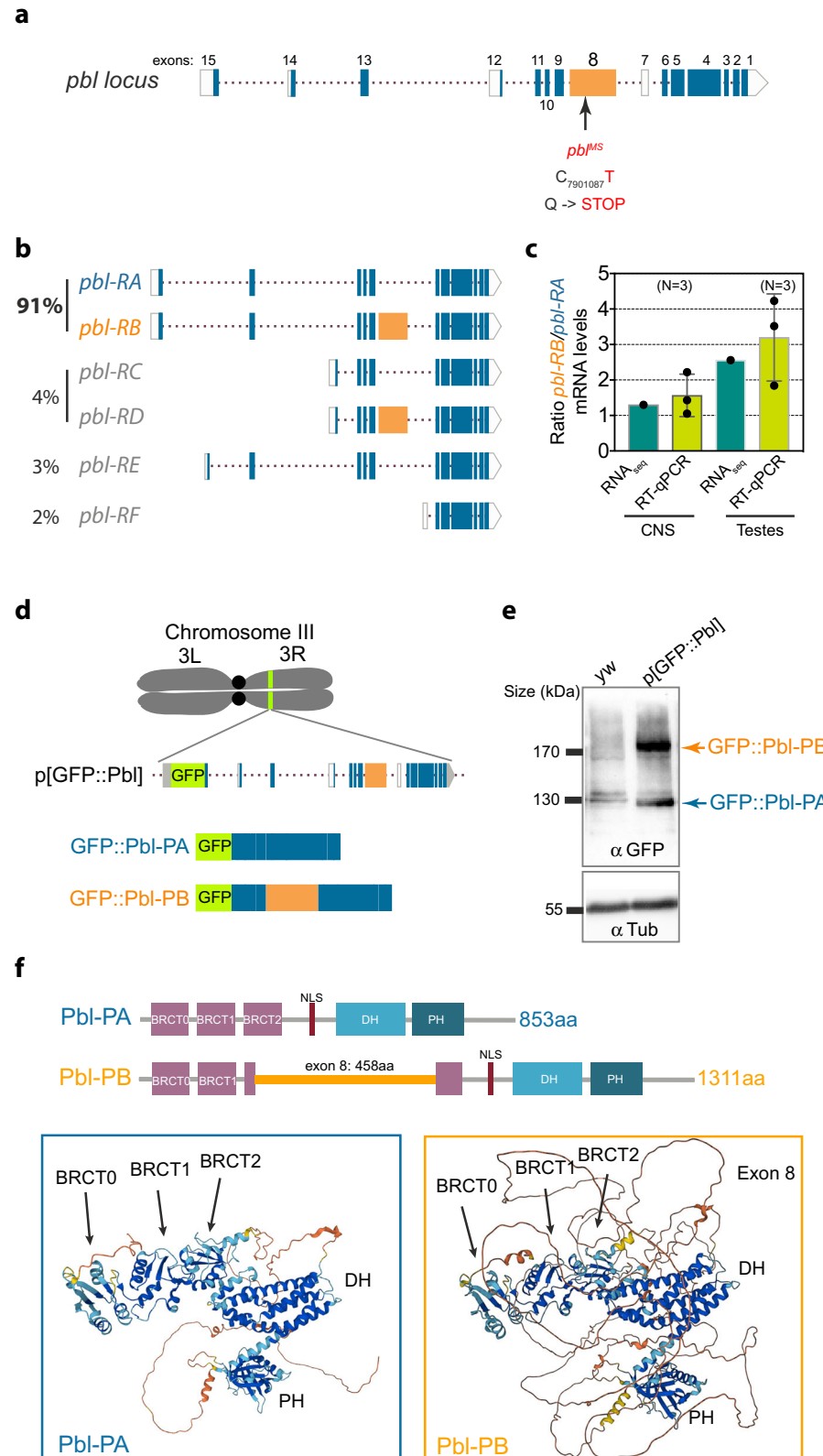

concentrates at the furrow of dividing embryonic cells but is not detected at the midzone of the central spindle[5]. In contrast to embryonic cells, in dividing neuroblasts, GFP::Pbl-A was detected at the midzone and furrow, concomitantly with its localisation at the polar membrane (Fig. 3a–e, Supplementary Movie 2). Intriguingly, it accumulated more prominently at the neuroblast than the GMC membrane (Fig. 3a and e). Upon completion of furrow ingression,

GFP::Pbl-A labelled the midbody and polar membrane of the nascent cells (Fig. 3a, f–i). GFP::Pbl-A was detected in the nucleus after completion of cleavage furrow ingression, as reported previously, and was both in the nucleus and plasma membrane in interphase cells (Fig. 3f–j)[36]. While GFP::Pbl-B shared a similar localisation at the polar membrane as GFP::Pbl-A during furrow ingression, striking differences in its dynamics were also observed. First, GFP::Pbl-B was neither

**Fig. 2 | pblMS is a non-sense substitution in an alternatively spliced exon specific to the Pbl-B isoform, which is expressed at equal levels as Pbl-A in the central nervous system. a** Diagram of the *pbl* gene locus. Filled boxes represent coding regions (exon 8 is highlighted in orange) and empty boxes correspond to potential untranslated sequence. The arrow and number indicate the base substitution (C to T) in exon 8 and the position on the chromosome. This non-sense mutation generates a glutamine (Q) to stop codon in the alternatively spliced exon 8. **b** Diagrams of the six predicted *pbl* isoform mRNAs from alternative splicing. The numbers represent the estimated frequencies of the pbl isoforms based on exon junction frequencies recovered from RNA-seq experiments. The isoforms pbl-RA and pbl-RB account for 91% of expressed messengers at that locus (number of reads in Supplementary Table 1). **c** Graph showing the Pbl-RB/Pbl-RA mRNA level ratio using exon junction analysis from central nervous system (CNS) and testes dataset (number of reads in Supplementary Table 1) as well as semi-quantitative RT-qPCR

from CNS and testes mRNA extracts. N, number of experiments. Bars indicate mean ± SD. Symbols indicate individual experiments. **d** Scheme of the p[pblGFP::Pbl] genomic construct inserted on the right arm of chromosome 3 used for the western blot shown in (**e**). **e** Western blot probed with anti-GFP antibodies to detect GFP::Pbl-A and GFP::B from larval brains of the indicated genotypes (top panel). Detection of α-tubulin was used as a loading control (bottom panel) (The experiment was repeated 3 times). **f** Diagrams and AlphaFold structure predictions of Pbl-PA and Pbl-PB. Pbl-A and B contain two BRCT domains in their N-termini and PH and DH domains in their C-termini. Pbl-PA possesses a third BRCT domain, BRCT2 in its N-terminus. The additional 458 amino acids of exon 8 are positioned within the third presumptive BRCT2 domain in Pbl-PB. However, the AlphaFold model predicts that the exon 8 amino acid sequence forms an unstructured loop that does not affect the folding of the third BRCT. Colour code of AlphaFold model confidence: dark blue, very high, light blue, high, yellow, low, orange, very low.

detected at the midzone nor at the midbody and was not enriched at the furrow. Second, it accumulated in the nucleus before completion of cleavage furrow ingression, prior to Pbl-A nuclear localisation, and was strictly nuclear at the end of cytokinesis and in interphase (Fig. 3a, f–i, Supplementary Movie 2). Importantly, GFP::Pbl-B dynamics were similar in the presence or absence of endogenous Pbl indicating that the absence of Pbl-B at the midzone is not due to the titration of RacGAP50C by endogenous Pbl-A (Fig. 3c–e and g–l, light and dark dots correspond to the presence or absence of endogenous *pbl*). Collectively, these results indicate that the presence of exon 8 promotes premature Pbl-B nuclear localisation and either inhibits its midzone localisation or favours its broad membrane localisation and premature nuclear localisation at the expense of the midzone.

### Distinct cytokinesis defects in Pbl-A and Pbl-B expressing neuroblasts

The striking differences in Pbl-A and B localisation combined with the observation that myosin dynamics and cell size asymmetry during neuroblast divisions are affected in *pbl^MS* cells that lack a functional Pbl-B suggest that the two isoforms act differently to control Rho1 activity during cytokinesis. To examine the role of Pbl-A and B isoforms individually during neuroblast division, we adopted a genomic rescue approach. We took advantage of an existing pAcman plasmid containing a 17 kb sequence including the *pbl* genomic sequence and the corresponding transgenic flies containing the *pbl* full-length construct (referred hereafter as *Pbl-A + B*)[46]. We produced two modified versions of this pAcman plasmid and the corresponding transgenic flies where the constructs were integrated at the same locus as the full-length to minimise variation in transgene expression. In the first version, we introduced a non-sense substitution in exon 8 mimicking the *pbl^MS* mutation, and thus preventing the production of a functional Pbl-B (referred hereafter as *Pbl-A*). In the second version, the intron between exon 8 and 9 was deleted to prevent alternative splicing of exon 8 and thus promote the expression of Pbl-B but not Pbl-A (referred hereafter as *Pbl-B*) (Fig. 4a). All three constructs rescued the embryonic lethality of the null transheterozygote *pbl^3/pbl^2* mutant[47]. However, while *Pbl-A + B* and *Pbl-A* produced viable adults at the expected frequency according to the principles of Mendelian inheritance, the frequency of *pbl^3/pbl^2* hatching adults expressing only Pbl-B was dramatically reduced (Fig. 4b). Next, we evaluated cytokinesis failure by quantifying the frequency of polyploid cells (Supplementary Fig. 5a, b). *Pbl-B* larval brains displayed a mild increase in the frequency of polyploid cells relative to *Pbl-A + B* or *Pbl-A*.

To assess the importance of each Pbl isoform on myosin dynamics during neuroblast cytokinesis, we monitored dividing cells expressing the different *pbl* constructs and the myosin marker Sqh::GFP. *pbl^3/pbl^2* neuroblasts expressing both Pbl-A and B (*Pbl-A + B*) exhibited myosin dynamics similar to WT cells, including myosin polar cortex enrichment at mid ingression (Fig. 4c–f, Supplementary Movie 3, compare

with Fig. 1a). Importantly, myosin remained prominently concentrated at the contractile ring, as illustrated by myosin intensity profiles from pole-to-pole and the ratio of cortex-to-furrow myosin fluorescence intensity (Fig. 4d, g, h). The cortical enrichment of myosin was associated with a slight elongation of the GMC, illustrating an increase in cortical contractility (Fig. 4i, j)[36]. The rate of ring constriction was comparable to WT cells (compare grey curves in Fig. 4k and Fig. 1f). Finally, the expression of both isoforms in *pbl* mutant neuroblasts gave similar GMC size as WT cells (Fig. 4l, m, compare the distribution of grey dots with WT distribution in Fig. 1h).

Next, we examined myosin dynamics in *pbl^3/pbl^2* neuroblasts expressing Pbl-A but lacking a functional Pbl-B. We found that myosin exhibited dynamics reminiscent of those observed in *pbl^MS* mutant cells. Myosin remained highly focused at the contractile ring throughout furrow ingression (Fig. 4c, d, Supplementary Movie 3). No polar cortical enrichment of myosin was detected (Fig. 4e–h). Moreover, while the ring constricted efficiently (Fig. 4k), the GMC cell size asymmetry was not preserved. Indeed, at the completion of cytokinesis, the GMCs were significantly smaller in cells expressing only Pbl-A than in cells expressing both isoforms (Fig. 4l, m). These results, first, confirm that the non-sense mutation identified in the *pbl^MS* sequence is responsible for the perturbed myosin dynamics and the abnormal cell size asymmetry observed in the dividing neuroblasts of the *pbl^MS* mutant. Second, they indicate that Pbl-B is required for the enrichment of myosin at the polar cortex. Third, they suggest a role for polar cortex contractility in maintaining ring position during furrow ingression, and more generally in preserving cell size asymmetry.

Next, we monitored myosin dynamics in *pbl^3/pbl^2* neuroblasts expressing only Pbl-B and, thus, lacking a functional Pbl-A. Within two minutes after anaphase onset, myosin depleted the poles and enriched the baso-equatorial cortex during contractile ring assembly, as observed for cells expressing only Pbl-A, or both isoforms. However, shortly after the onset of furrow ingression, myosin displayed a distinct pattern. Myosin spread rapidly toward the polar cortex of both nascent cells (Fig. 4c–f, Supplementary Movie 3). This was concomitant with its dilution from the contractile ring as illustrated by the lack of maximum levels of myosin at the furrow in the myosin intensity linescan from pole-to-pole and the increase in the ratio of cortex-to-furrow myosin levels (Fig. 4d–h). This was accompanied by severe widening of the furrow, which, in some instances, adopted an extreme tubular shape (Fig. 4c, i, j, Supplementary Movie 3). In most *Pbl-B* cells, ring constriction was not as efficient as in *Pbl-A* cells (Fig. 4k). Despite these altered myosin dynamics and the slower rate of ring constriction, the GMC cell size was preserved (Fig. 4l, m). This wide furrow in *Pbl-B* cells is reminiscent of the enlarged contractile ring observed when trailing chromatids are present at the cleavage site in wild type cells[36,37]. To test the possibility that the phenotypes observed in *Pbl-B* cells are due to lagging chromosomes, we examined chromosome segregation concomitantly with myosin dynamics in this genetic

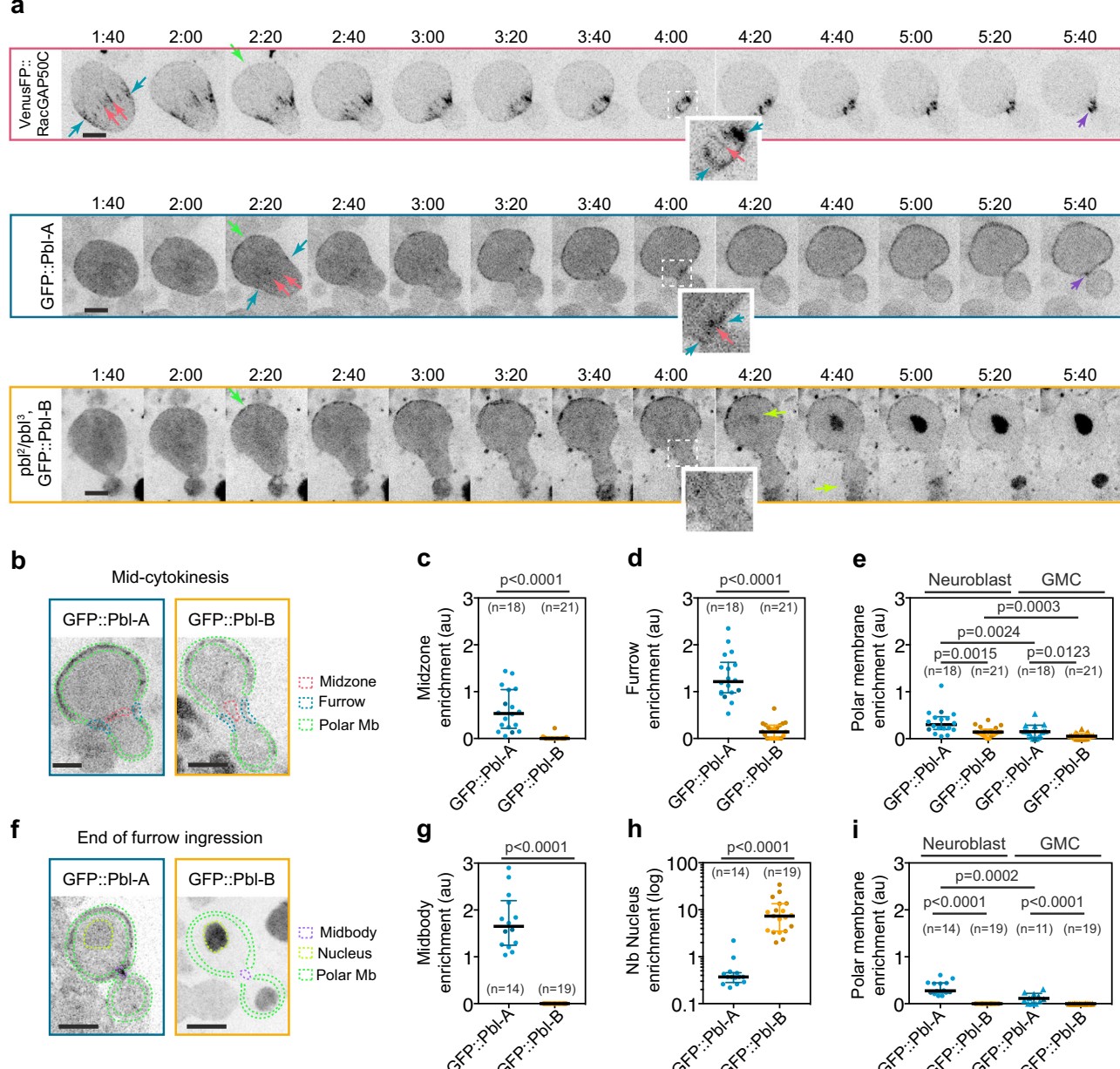

**Fig. 3 | GFP::Pbl-A and GFP::Pbl-B have distinct localisation patterns during cytokinesis. a** Time-lapse images of neuroblasts of the indicated genotype during cytokinesis. The green, blue, pink, yellow and purple arrows indicate the detection of the proteins at the plasma membrane, furrow, midzone, nucleus and midbody respectively. Insets are magnified images of the midzone and furrow region (indicated by the dashed square in the image). Time, min:sec, from anaphase onset. (number of cells, *n* = 28, 18 and 21 for VenusFP::RacGAP50C, GFP::Pbl-A and GFP::Pbl-B, respectively). **b** Sagittal images of neuroblasts of the indicated genotype at mid-cytokinesis. The mean fluorescence intensity of GFP::Pbl-A and B were measured at the midzone (pink dashed area), furrow (blue dashed area) and polar membrane (green dashed area) and plotted in (**c**), (**d**) and (**e**), respectively. **f** Sagittal images of neuroblasts of the indicated genotype at the end of furrow ingression. The mean fluorescence intensity of GFP::Pbl-A and B were measured at the midbody (purple dashed area), the nucleus (yellow dashed area) and the polar membrane (green dashed area) and plotted in (**g**), (**h**) and (**i**), respectively. In all scatter dot plots, dark colour dots correspond to *pbl³/pbl²* genotype and bars correspond to median ± interquartile range. n number of cells. A Mann–Whitney two-tailed test was used to calculate *P* values. Scale bar, 5 µm for all images.

background (Supplementary Fig. 6). No chromosome segregation defects were detected in *Pbl-B* cells exhibiting a transient widening of the furrow associated with excessive myosin enrichment at the polar cortex (Supplementary Fig. 6a, b).

### RacGAP50C depleted neuroblasts exhibit similar cytokinesis defects as cells lacking Pbl-A

Changes in myosin dynamics, the rate of furrow ingression and the altered GMC cell size between *Pbl-A* or *Pbl-B* cells could result from

the difference in Pbl-A and B localisation, thus inducing distinct patterns of Rho1 activation. Alternatively, they could result from a difference in Pbl-A and B function regardless of their localisation. To distinguish between these possibilities, we sought to prevent Pbl-A midzone localisation and examine the outcome in terms of myosin dynamics, furrow ingression and daughter cell size asymmetry. Previous studies have shown that the depletion of RacGAP50C prevents Pbl-A localisation at the furrow during cytokinesis in embryonic cells[19]. Similarly, MgcRacGAP is essential for Ect2 localisation to the

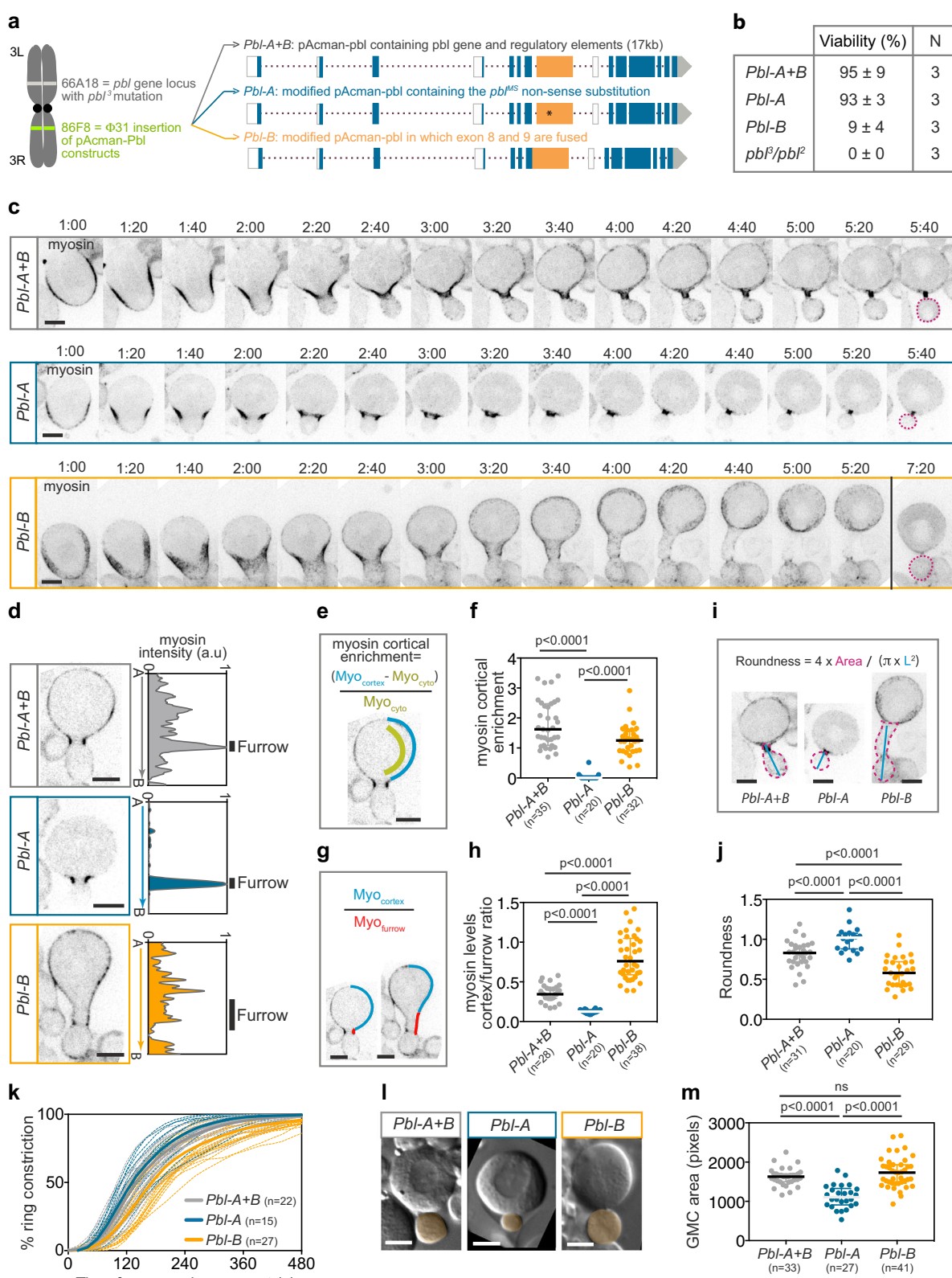

midzone of the central spindle in mammals[17]. We therefore, first, determined if the Pbl-A midzone localisation in neuroblasts was dependent on RacGAP50C.

Depletion of RacGAP50C by RNAi specifically in imaginal discs and the central nervous system during larval development induced massive death at the pupal stage and no survival to adulthood. In addition, a large number of polyploid cells were observed in third instar larval brains as previously reported, illustrating the efficiency of the RNAi used to deplete RacGAP50C (Supplementary Fig. 7a, b)[29].

Next, we monitored GFP::Pbl-A and B localisation in neuroblasts depleted for RacGAP50C. We found that, while the broad localisation of GFP::Pbl-A at the nascent cell plasma membrane was not affected, no signal was detected at the midzone and no enrichment was observed at the site of furrowing (Fig. 5a). In addition, GFP::Pbl-A nuclear

**Fig. 4 | Pbl-A and B are both required for robust asymmetric neuroblast division. a** Scheme of chromosome III showing the location of the *pbl* gene locus and the Φ31 insertion of the pAcman-pbl genomic constructs indicated on the right. The construct "Pbl-A + B" corresponds to the *pbl* gene allowing the expression of both Pbl-A and B[46]. "Pbl-A" construct corresponds to *pbl* gene carrying the nonsense substitution identified in *pbl^MS*. This allows the expression of Pbl-A but prevents the expression of a functional Pbl-B. "Pbl-B" construct corresponds to *pbl* gene where exon 8 and 9 are fused to allow the expression of Pbl-B but prevent the expression of Pbl-A. These pAcman insertions were recombined with the *pbl^3* null mutation. **b** Percentage of survival to adulthood of *pbl^3/pbl^2* expressing the indicated Pbl construct. **c** Time-lapse images of *pbl^3/pbl^2* neuroblasts expressing the indicated Pbl constructs and Sqh::GFP (myosin). Time, min:sec from anaphase onset. (number of cells, *n* = 35, 20 and 32 for *Pbl-A* + *B*, *Pbl-A* and *Pbl-B* respectively). **d** Sagittal images of neuroblasts and the corresponding cortical myosin intensity measurement from apical (A) to basal (B) poles. The black vertical lines represent the position and width of the furrow on the line-scan. (number of cells, *n* = 35, 20

and 32 for *Pbl-A* + *B*, *Pbl-A* and *Pbl-B*, respectively). **e** Scheme illustrating the method to quantify myosin cortical enrichment plotted in (**f**). **f** Scatter dot plot of cortical myosin enrichment. **g** Scheme illustrating the method to calculate the ratio of cortex-to-furrow myosin levels plotted in (**h**). The blue and red lines correspond to the cortex and furrow, respectively. **h** Scatter dot plot of cortex-to-furrow myosin level ratios. **i** Selected images of neuroblasts of the indicated genotypes to illustrate the measurement of the area (pink dashed line) and the length (blue line) of the presumptive GMC for the calculation of the roundness plotted in (**j**). **j** Distribution of presumptive GMC roundness for the indicated genotypes. **k** Percentage of ring constriction over time for the indicated genotypes from anaphase onset to mid-body stage (100% constriction). Dashed and solid lines correspond to individual cell and average, respectively. **l** DIC images of the indicated genotypes at the end of cytokinesis. The GMC area is highlighted in orange. **m** Distribution of GMC areas for the indicated genotypes. Bars represent median ± interquartile range. n number of cells. A Mann–Whitney two-tailed test was used to calculate *P* values. Scale bars, 5 μm.

---

localisation was not accelerated upon depletion of RacGAP50C. GFP::Pbl-B exhibited similar dynamics in RacGAP50C depleted cells as in WT cells, including its nuclear localisation prior to Pbl-A (Fig. 5a). We conclude that RacGAP50C is required for Pbl-A midzone and furrow localisation, but is not essential for its recruitment to the plasma membrane. In addition, this data indicates that exon 8 protein sequence accelerates Pbl-B nuclear localisation. Remarkably, these RacGAP50C depleted cells exhibited transient enlarged furrows reminiscent of those observed in cells expressing Pbl-B.

These results prompted us to examine myosin dynamics, the rate of ring constriction and GMC cell size upon RacGAP50C depletion. Control cells exhibited myosin dynamics similar to those reported for wild type cells throughout cytokinesis (Fig. 5b–e, Supplementary Movie 4). In contrast, attenuation of RacGAP50C altered myosin dynamics in a similar way as cells expressing only Pbl-B. At mid-ingression, myosin formed an unusually large ring that produced an abnormally wide furrow (Fig. 5b, c, f, g and Supplementary Movie 4). Subsequently, myosin underwent extensive cortical enrichment as illustrated by the high ratio of cortex-to-furrow myosin levels compared to control cells (Fig. 5c–g). The broadening of myosin cortical localisation occurred concomitantly with a decrease in the rate of furrow ingression (Fig. 5h). These cytokinesis defects were associated with a mild proportion of larger GMCs than normal at the end of furrow ingression (Fig. 5j). Since RacGAP50C depleted cells share similar phenotypes as *Pbl-B* cells, including excessive myosin cortical relocalisation, an enlarged furrow and a slower rate of furrow ingression, we conclude that these phenotypes are not the result of the absence of Pbl-A, but are rather the result of a lack of midzone and furrow-localised Pbl-A signalling which favours Rho1 activation through broad Pbl-B membrane localisation.

## Discussion

While the signalling pathway that specifies the zone of Rho1 activation to initiate contractile ring assembly is well defined, less is known about the mechanisms that control Rho1 activity to maintain ring position while sustaining efficient constriction. In this study, we found that Rho1 activation during asymmetric division of *Drosophila* neural stem cells relies on two Rho1 GEF isoforms displaying distinct localisation patterns. These two isoforms modulate Rho1 activity concurrently to promote robust furrow ingression while allowing the adjustment of its asymmetric position during ring closure. Based on our findings, we propose the following model: Pbl-A, predominantly concentrated at the midzone and furrow site through its association with RacGAP50C, maintains a threshold of Rho1 signalling at the leading edge of the furrow to sustain efficient ingression. After initiation of furrowing, the broad localisation of Pbl-B at the plasma membrane enlarges the zone of Rho1 activity, which, in turn, drives myosin efflux from the contractile ring and its enrichment at the polar cortex. This increase in

pole contractility contributes to the fine-tuning of furrow position to preserve daughter cell size asymmetry. When Pbl-B is absent, Pbl-A restricts Rho1 activity to a narrow zone, which is sufficient to drive efficient ring constriction but is insufficient to adjust furrow position. Consequently, abnormally small GMCs are produced, depending on the initial position of the furrow. When Pbl-A is not enriched at the midzone and furrow, Pbl-B-dependent broadening of Rho1 activity dilutes active myosin from the edge of the furrow by promoting its excessive enrichment at the polar cortex. As a consequence, the furrow enlarges, temporarily diminishing its rate of ingression (Fig. 6).

The identification of the *pbl^MS* mutation as a non-sense substitution in the alternatively spliced exon 8 led us to uncover Pbl-B as one of the two major Pbl protein isoforms expressed in *Drosophila* tissues. The premature stop codon in exon 8 prevents the expression of a functional Pbl-B in the *pbl^MS* mutant. Since mRNA with premature stop codons are often rapidly degraded by the nonsense-mediated mRNA decay (NMD) pathway[48,49], the *pbl^MS* phenotypes observed during neuroblast divisions are likely the result of a loss of Pbl-B function rather than the dominant effect of truncated Pbl-B expression. However, we cannot formally exclude the latter possibility.

No role has been assigned to the Pbl-B isoform prior to our work. Two distinct functions for Pbl during embryonic development have been identified through genetic screens: (1) its function as a GEF for Rho1 during cytokinesis[5,14–16] and (2) its Rho1-independent role in the epithelial to mesenchymal transition (EMT) and mesoderm cell migration[46,50–54]. The fact that the expression of Pbl-B rescues the embryonic lethality of *pbl* null mutants and produces adult escapers indicates that it contributes to some extent redundantly with Pbl-A to perform these two independent functions.

Structural modelling predicts that Pbl-B, like Pbl-A, may contain three folded BRCTs at its N terminus despite the presence of 458 extra amino acids within the third BRCT sequence. Since BRCTs are critical for Pbl-A interaction with RacGAP50C, this implies that Pbl-B may bind RacGAP50C. However, several lines of evidence suggest that, in vivo, Pbl-B functions independently of its interaction with RacGAP50C. First, Pbl-B is not detected at the spindle midzone and is not concentrated at the furrow, whereas Pbl-A localises to the midzone and furrow site in a RacGAP50C-dependent manner. Second, expression of only the Pbl-B isoform produces phenotypes, including an enlarged furrow and slower rate of ingression, akin to that observed in cells depleted for RacGAP50C. Hence, we propose that two parallel pathways, a RacGAP50C-dependent Pbl-A and a RacGAP50C-independent Pbl-B pathway act in concert to promote robust asymmetric division in neuroblasts. Consistently, studies in *C. elegans* one-cell embryos revealed that two parallel pathways, a Cyk-4-dependent and independent pathway, drive furrow ingression[55,56]. In mammalian cells, the association of Ect2 with the plasma membrane or the local activation of RhoA through the use of optogenetics is sufficient to induce

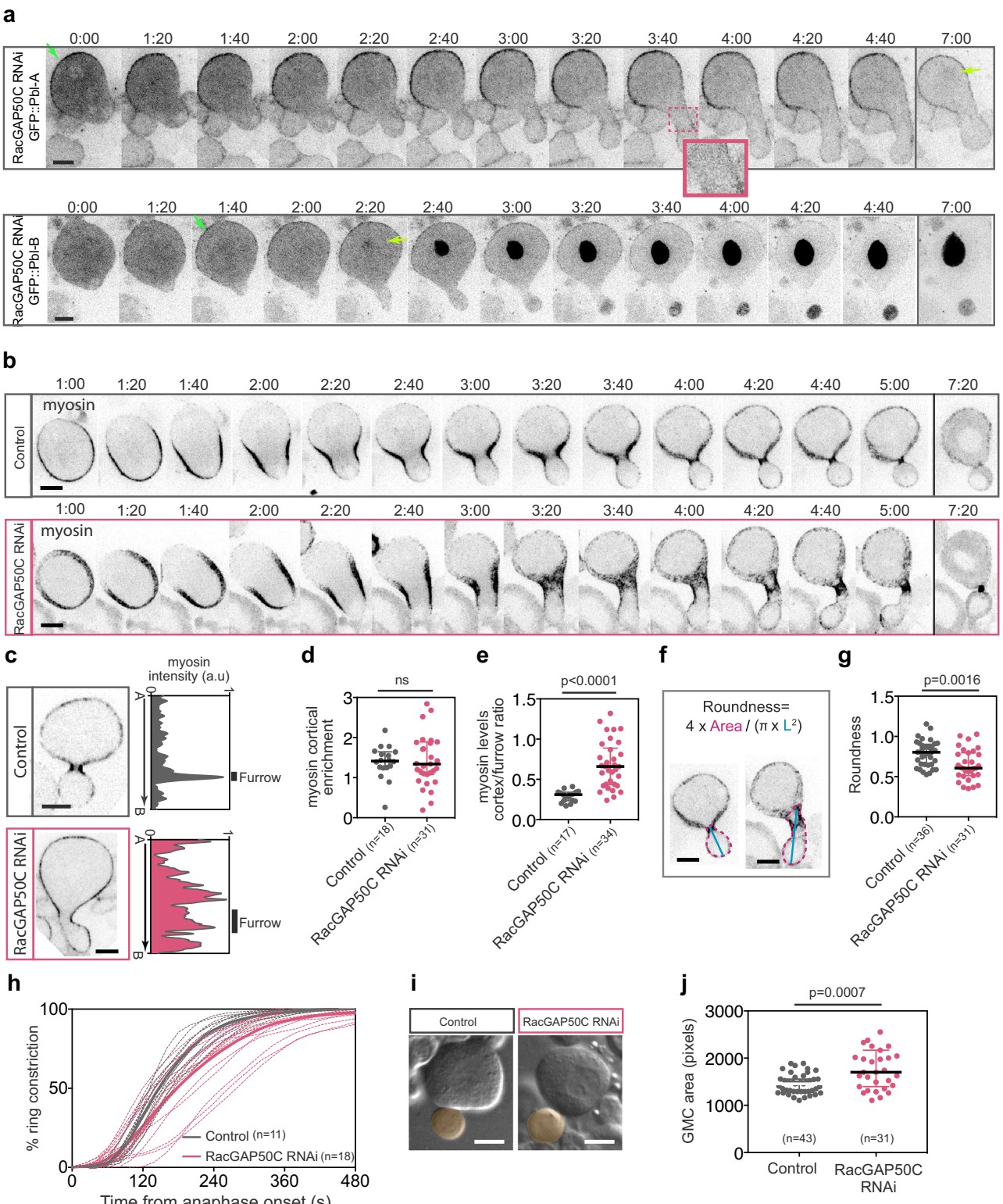

furrowing[9,57]. Remarkably, successful cytokinesis occurs in cells expressing Ect2 mutated on residues that impairs its interaction with MgcRacGAP and prevents its recruitment to the spindle midzone, highlighting an MgcRacGAP-independent redundant pathway for Ect2 activation[57]. Recently, a novel isoform of human Ect2 has been identified through screens for human transformed cells resistant to treatment with Doxorubicin and Paclitaxel[58,59]. Importantly, this long human Ect2 isoform is strictly nuclear in interphase like Pbl-B and, also reminiscent of Pbl-B, contains additional amino acids in one of the

BRCT domains that may affect its folding and impair Ect2 interaction with MgcRacGAP. Thus, the use of redundant RacGAP-dependent and independent pathways for Ect2/Pbl activation during cytokinesis may be a convergent mechanism in multicellular organisms, making a process that has to adapt to a variety of cell types with specific extrinsic and intrinsic physical constraints more robust.

The model that Pbl-B drives cytokinesis independently of Rac-GAP50C raises the question of how the furrow site is specified in these cells. Previous studies have identified two distinct pathways that define

**Fig. 5 | Cytokinesis defects in RacGAP50C depleted neuroblasts are similar to that observed in cells lacking Pbl-A. a** Time-lapse images of neuroblasts depleted of RacGAP50C by RNAi and expressing GFP::Pbl-A or GFP::Pbl-B. Inset shows the magnified midzone and furrow region delineated by the dashed pink square on the image. Time 0:00 corresponds to onset of furrow ingression (as defined as the onset of inward membrane curvature). Green and yellow arrows indicate polar membrane and nuclear localisation. (number of cells, *n* = 11 and 11 for Control and RacGAP50C RNAi, respectively). **b** Time-lapse images of control neuroblasts or neuroblasts depleted of RacGAP50C by RNAi (RacGAP50C RNAi) expressing Sqh::GFP (myosin) from early anaphase (1 min after anaphase onset) until the end of furrow ingression. Time 0:00 corresponds to anaphase onset. (number of cells, *n* = 18 and 34 for Control and RacGAP50C RNAi, respectively). **c** Sagittal images of neuroblasts of the indicated genotypes expressing Sqh::GFP (myosin) and the corresponding cortical myosin intensity measurement at the cortex from apical-to-basal poles. The vertical lines indicate the position and width of the furrow on the line-scan. **d** Scatter dot plot showing cortical myosin enrichment and (**e**) the ratio of cortex-to-furrow myosin levels for the indicated genotypes. **f** Selected images of neuroblasts to illustrate the area (pink dashed line) and the length (blue line) measured for the calculation of the roundness of the presumptive GMC plotted in (**g**). **g** Scatter dot plot showing the roundness of the presumptive GMC for the indicated genotype at the time of maximum deformation. **h** Graph showing the percentage of ring constriction over time for the indicated genotypes from anaphase onset to midbody stage (100% constriction). Dashed and solid lines correspond to individual cell and average, respectively. **i** Selected DIC images from time-lapse of neuroblasts of the indicated genotypes at the end of cytokinesis. The GMC area is highlighted in orange. **j** Scatter dot plot showing the GMC area for the indicated genotypes. Bars represent median ± interquartile range for all scatter dot plots. n, number of cells. A Mann–Whitney two-tailed test was used to calculate *P* values. Scale bars, 5 μm. Time, min:sec.

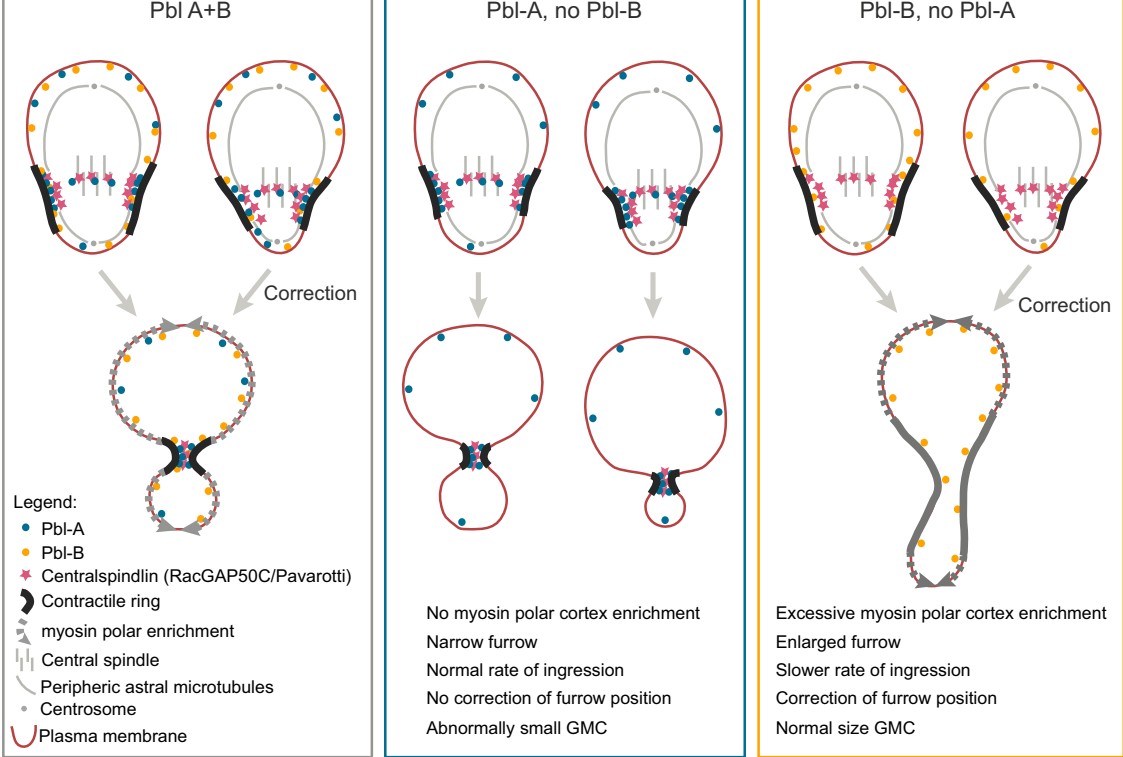

**Fig. 6 | Model for Pbl-A and B function during neuroblast asymmetric division.** Pbl-A (blue circle) remains predominantly at the midzone and site of furrowing through its association with the centralspindlin component RacGAP50C (pink star), which interacts with two pools of microtubules, the peripheral astral microtubules and the central spindle. The enrichment of Pbl-A at the furrow sustains a threshold of Rho1 activity to promote efficient ring constriction. Pbl-B localises broadly on the plasma membrane where it enlarges the zone of Rho1 activity throughout the second phase of furrow ingression. This promotes myosin polar cortex enrichment on both nascent daughter cells. The balance between the activities of these two Pbl-isoforms allows efficient furrow ingression while adjusting contractile ring position to generate appropriately-sized daughter cells. In cells expressing Pbl-A but not Pbl-B, Rho1 activity remains focused at the furrow throughout ingression. As a result, myosin does not enrich the polar cortex. This focused Rho1 activity permits efficient ring constriction but prevents adjustment of furrow position. Consequently, abnormally small GMCs are produced in the instance where the furrow is initially too basally positioned. In cells expressing Pbl-B but not A, Pbl-B broad membrane localisation promotes Rho1 activation in a larger zone after furrow initiation. Consequently, myosin undergoes excessive polar cortex enrichment. This induces the transient widening of the furrow accompanied by a decrease in its rate of ingression. Normal sized GMC are nevertheless produced at the end of furrowing.

the future cleavage site in *Drosophila* neuroblasts. One pathway involving the Pins polarity determinant complex acts early to promote myosin depletion from the apical pole and accumulation at the basal cortex[28,30,32,33]. The second pathway depends on the microtubules-associated centralspindlin complex that acts to stabilise the baso-equatorial position of the contractile ring[29–31]. Since myosin dynamics during furrow site specification are unperturbed in cells lacking Pbl-A or depleted for RacGAP50C, we propose that the polarity-determinant pathway may be the predominant pathway that positions the cleavage site in these cells. The target of the polarity pathway for cleavage site specification is currently unknown. It would be interesting in future to determine if this pathway acts through Pbl-B regulation.

One of our most striking findings was that cells without Pbl-B produce aberrantly small GMCs. The observation that GMC size is correlated with the initial position of the furrow in these cells suggests a defect in furrow position correction. Previous studies determined that two pools of the centralspindlin complex separated by two different populations of microtubules (the central spindle and the peripheral astral microtubules) competitively influence furrow position during ingression[31]. In this study, perturbation of the peripheral

microtubule-associated pool of centralspindlin produced larger GMCs than normal. Since our data suggest that Pbl-B acts independently of the centralspindlin complex and that cells lacking Pbl-B produce GMCs that are smaller than normal, we propose that an additional mechanism to the centralspindlin-dependent pathway participates in furrow position adjustment. This mechanism is likely to involve the change in mechanical properties of the nascent daughter cell cortices as active myosin enriches the cortex during ring constriction. Indeed, myosin enrichment at the polar cortex coincides with an increase in contractility, as illustrated by the transient deformation of the nascent cells (Fig. 4j). Supporting this idea, studies in mammalian cells have demonstrated that perturbation of the contractile properties of one nascent daughter cell polar cortex induces furrow displacement associated with cell shape instability[60,61]. Additionally, two myosin isoforms with different motor kinetics participate in polar cortex contractility, which is critical for maintaining cell shape stability and promoting the fidelity of cytokinesis[62,63].

We provide the ground-work for future studies on how changes in the mechanical properties of the polar cortex contribute to adjust furrow position during asymmetric stem cell divisions. It will be equally interesting to assess this question in other cell types and in relation to tissue features. Pbl-A and B isoforms provide excellent genetic tools to address these questions as they allow the physiological modulation of Rho1 activity at the furrow and polar membrane.

## Methods
### Fly stocks
Flies were raised on standard media at 25 °C. The *pbl²* and *pbl³* alleles were described previously[5,16,47]. The screen that identified *pblᴹˢ* (also called pblZ4836) was described in ref. [38]. P[UAS > RacGAP50C-dsRNA] (stock n°6439) and P[69B > Gal4] strains were provided by Bloomington stock centre (Indiana, USA). P[w+, sqh > Sqh::GFP transgenic strain was described in refs. [64,65]. P[UAS > VenusFP::RacGAP50C] was described in ref. [18]. Pbl-A + B stock was described in ref. [46]. Pbl-A, Pbl-B, GFP::Pbl-A and GFP::Pbl-B transgenic stock were produced in our laboratory for this study (see Supplementary Table 2 for the genotype of flies used in this study).

### Sequencing of pebble genomic region
Genomic DNA was extracted from 30 adult flies previously collected and frozen at −80 °C, by grinding in 400 µL of buffer containing 100 mM Tris-HCl pH 7.5, 100 mM EDTA, 100 mM NaCl, 0.5% SDS, until only cuticles remain. The mixture was incubated for 30 min at 65 °C. Then 800 µL of a solution of LiCl/KAc (1 part of 5 M KAc: 2.5 parts 6 M LiCl) was added and the mixture incubated on ice for 10 min. After centrifugation at 12,000 g for 15 min, 1 mL of supernatant was transferred to 600 µL of isopropanol. After 15 min of centrifugation at 12,000 g, the pellet containing the genomic DNA was washed with 70% ethanol and dried before resuspension in 150 µL of diH2O.

Amplification by PCR of three fragments covering the whole genomic region of pebble was performed using oligonucleotide pairs A to C from genomic DNA of WT and homozygous *pblᴹˢ* flies (Supplementary Table 3). Each fragment was sequenced for complete coverage of exons in the pebble gene. Sequences were analysed with Serial Cloner software and compared to Flybase annotated genome (FBgn0003041).

### Transcriptomic analysis and RT-qPCR
The identification of *pbl* exon junction usage was performed as described in the result section and in ref. [40]. To estimate the level of pebble mRNA isoforms in larval central nervous systems (CNS) and adult testes, total RNAs from 60 larval CNS or 80 adult testes were extracted using RNAspin Mini kit (GE Heathcare). 1 µg of total RNAs of each extract were converted into cDNA using oligodT and SuperScript III Reverse transcriptase (Invitrogen) according to the manufacturer's

protocol. Quantitative PCR was performed using SSO Advanced Universal SYBR Green Supermix (Bio-Rad).

### Cloning
Pbl-A and B constructs were obtained by DNA synthesis using pAcman[Pbl] plasmid described previously[46]. For Pbl-A, the cytosine at position 1114 (Q372) was substituted to thymidine to mimic the *pblᴹˢ* mutation. For Pbl-B, the intron sequence between exons 8 and 9 was removed (Genebridges). GFP::Pbl-A and GFP::Pbl-B were cloned as follows: Pbl-A and B cDNAs produced by RT-PCR from adult mRNA extracts were inserted into pAcman[GFP] using AscI and NotI sites flanking a cassette obtained by PCR and containing sequences as follows: AscI-pbl promoter-pbl5'UTR-GFP-pbl (A or B) cDNA -pbl3'UTR-NotI. The resulting plasmids pAcman[Pbl-A], pAcman[Pbl-B], pAcman[GFP::Pbl-A] and pAcman[GFP::Pbl-B] were used to produce transgenic flies with the insertion at the same landing site, 86F8b on the third chromosome. The plasmid pAcman[GFP::Pbl-A + B] was produced by inserting the eGFP sequence before the ATG of the genomic Pbl sequence present in pAcman[Pbl] using AscI and BsiW restriction enzymes. The resulting pAcman[GFP::Pbl] plasmid was inserted at the landing site 68A4 on the third chromosome of the fly genome.

### Western-blotting
60 brains from 3rd instar larvae of each genotype were dissected in 1X PBS and transferred in 30 µL of RIPA buffer (Tris-HCl 10 mM pH 7.5, NaCl 150 mM, EDTA 0.5 mM, SDS 0.1%, Triton 1%, Na Deoxycholate 1%) supplemented with protease inhibitor cocktail (Roche) and PMSF (1 mM). After grinding and homogenization with a pestle in a 1.5 mL centrifuge tube, lysates were centrifuged at 15,000 g for 5 min at 4 °C and supernatants containing proteins were transferred into equal volume of Laemmli 2X buffer. Boiled samples were loaded and analysed by 10% SDS-PAGE. Proteins were transferred to nitocellulose membrane using a semi-dry electrophoretic blotting device (BioRad). Mouse anti-GFP (Roche, clones 7.1 and 13.1, 2 µg/ml) and mouse anti-αTubulin primary antibodies (Sigma, clone DM1A, 2 µg/ml) and anti-mouse HRP conjugated secondary antibodies (DakoCytomation, P0217, 2 µg/ml) were used for western blotting.

### Microscopy
Live imaging of larval neuroblasts was carried out as follows: late third instar larval CNS were dissected in 1x PBS and transferred to a drop of 1x PBS on a 22 × 22 mm glass coverslip. A glass slide (60 × 24 mm) was gently placed on top of the coverslip until the coverslip adhered to the slide by capillary action. The CNS was slightly squashed by absorption of excess liquid from the edges of the coverslip to facilitate visualisation of dissociated neuroblasts. The coverslip was then sealed with halocarbon oil 700 (Sigma)[66]. The preparation was kept for a maximum of one hour. In this study, neuroblasts above 11 µm in diameter were observed except for Fig. 3b–i where smaller neuroblasts were also analysed. Live analysis was performed at room temperature with 100X oil Plan-Apochromat objective lens (NA 1.4) and an Axio-Observer.Z1 microscope (Carl Zeiss) equipped with a spinning disk confocal CSU-X1 (Yokogawa), an EMCCD Evolve camera (Photometrics). The fluorescent proteins were excited with a 491 nm (100 mW; Cobolt Calypso) and 561 nm (100 mW; Cobolt Jive) lasers. Metamorph software (Molecular Devices) was used to acquire the data. Images were acquired every 20 s over 7 µm depth with a z step of 0.5 µm. DIC images and the relevant fluorescent images were acquired near-simultaneously for all time points. Since chromosomes are readily detected by DIC illumination, we used anaphase onset, defined as initiation of sister chromatid separation, as the time reference (time, 0:00 min:sec) in all our figures except Fig. 5a, where time, 0:00 refers to the onset of furrow ingression. Images are maximum projections of the relevant z unless specified in the figure legend. For all genotypes including RacGAP50C RNAi, diploid neuroblasts were observed. Diploid neuroblasts are readily distinguished from their

polyploid counterparts by their diameter at metaphase and the size of the chromosome mass.

## Image analysis and data processing

Fiji (National Institute of Health) was used for image analysis and fluorescence quantification[67]. The background level of the camera was subtracted from all raw data before quantification. All fluorescence quantification was performed on a single plane of the relevant Z position. For the line-scan in Figs. 1c, 4d and 5c, a 2-pixel wide line was drawn around the cortex from one cell pole to the other at the time of maximum myosin polar cortical enrichment (-80% furrow ingression). The average grey value in the cytoplasm was subtracted from the data. The maximum grey value was used for normalisation and the line-scan was plotted (6 neighbours average) using Prism (GraphPad). For Figs. 1e, 4f and 5d, the average grey value of myosin at the neuroblast cortex ($Myo_{cortex}$) and the cytoplasm ($Myo_{cyto}$) was measured using a 2-pixel wide line as illustrated in the schemes (Figs. 1d and 4e). Myosin cortical enrichment was calculated as follows: ($Myo_{cortex}$- $Myo_{cyto}$)/ $Myo_{cyto}$. A similar method was used to measure GFP::Pbl-A and B polar membrane and furrow enrichment shown in Fig. 3d, e, i. For Fig. 3c, g, h, the mean grey value of the relevant ROI (as shown in Fig. 3b, f) was measured and the enrichment calculated as follows: (ROI-cyto)/cyto. Measurement of GFP::Pbl-A and B signals were performed at two time points during cytokinesis, mid-cytokinesis and at the end of furrow ingression as illustrated in Fig. 3b, f. For Figs. 4h and 5e, the mean grey value of myosin at the furrow ($Myo_{furrow}$) was measured by drawing a 2-pixel wide line at the furrow as indicated in the scheme in Fig. 4g. The length of the furrow was determined by the divergent angle from the long axis of the cell being approximately <30°. The ratio $Myo_{cortex}$/ $Myo_{furrow}$ was then calculated. For Fig. 1i, the time of initiation of furrow ingression was determined as the time of inward membrane curvature. The position of the furrow was defined as the maximum point of curvature. The percentage of ring ingression plotted in Figs. 1f, 4k and 5h were measured as indicated in the scheme in Fig. 1f. The constriction was considered 0% at anaphase onset and 100% when the ring reached midbody size (3 to 5 pixels in diameter). Individual cell and average measurements were plotted using Prism (4 neighbours average)(GraphPad). The GMC area plotted in Figs. 1h, 4m and 5j were quantified as follows: an ROI delineating the GMC on a single z image of the most GMC sagittal view using the Sqh::GFP signal was drawn and the number of pixels within the ROI determined. The accuracy of the ROI was confirmed using the DIC channel.

## Viability analysis

For viability assays, flies were raised on standard media at 25 °C. Eight young males $pbl^2$/TM6B were crossed with ten young females of the following genotypes: $pbl^3$, pAcman[Pbl-A + B]/TM6B, $pbl^3$, pAcman[Pbl-A]/TM6B, $pbl^3$, pAcman[Pbl-B]/TM6B or $pbl^3$/TM6B. For each genotype, the parents were left to lay eggs in a vial every 2 days for 8 days. The first vial was discarded and all the adult progeny of the remaining vials were counted. The percentage of viability of the transheterozygous individuals carrying the pAcman Pbl constructs was determined by calculating the ratio of $pbl$ transheterozygote vs TM6B flies over the expected frequency of transheterozygotes according to Mendel's law of inheritance. The experiment was performed three times independently.

## Cytology

Third instar larval brain were dissected in 1X PBS, fixed 10 s in 45% followed by 10 s in 60% acetic acid solutions and squashed between slide and coverslip in an aceto-orcein solution to stain the chromosomes[68]. To quantify diploid and polyploid mitotic cells, preparations were observed by phase contrast using a Nikon eclipse Ti equipped with 100X objective.

## Sequence alignement of Pbl orthologs

948 protein sequences of Pbl orthologs were retrieved from the OrthoDB v11 database (https://www.orthodb.org/?query=LOC4812 378;[42]) and aligned with the sequence of the Pbl-B isoform of *D. mela-nogaster*. Sequences were aligned using MAFTT 7.407[69] and run in "linsi" mode with options "–amino–reorder–treeout–localpair–maxiterate 1000". The alignment was visualised in AliView 1.26[70], ClustalX 2.1[71] and Seaview 5.0.5[72]. This file is available as supplementary file "pbl_Or-thoDB_eukaryota_MAFFT_alignment_reformatted.fasta.gz" in FASTA format.

## Reporting summary

Further information on research design is available in the Nature Portfolio Reporting Summary linked to this article.

## Data availability

Source data are provided for this paper. Source data are provided with this paper.

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

## Acknowledgements
We thank Andrew Murray for sharing reagents. We thank Régis Giet and Cameron Mackereth for helpful discussion and critical reading of the manuscript. We thank Marie Didelon for technical help and Gilles Hickson for insightful discussion. E.M. and L.B. were supported by Centre National de la Recherche Scientifique (CNRS). A.R. and D.McC. were supported by CNRS and the Agence Nationale de la Recherche (ANR-n°234520). A.R. was also supported by the European Research Council (ERC-GA311358) and the Conseil Régional d'Aquitaine (2014-1R30412-00003094). M.C.C. was supported by the University of Bordeaux. I.D. was supported by the Fondation pour la Recherche Médicale (FRM-ECO202106013724) and CNRS. N.J.T. and D.D. were supported by INSERM.

## Author contributions
E.M., I.D., M.C.C., L.B., N.J.T., D.D., D.McC. and A.R. performed experiments and data analysis and participated in the writing of the article.

## Competing interests
The authors declare no competing interests.
