## [Peer Review File · Nature Communications]

Two RhoGEF isoforms with distinct localisation control furrow position during asymmetric cell divisionReviewer #1 (Remarks to the Author):

This study sheds light on the function of different isoforms of the evolutionarily conserved RhoGEF Pebble in flies in the context of the regulation of physical size asymmetry between daughter cells in stem cells of the developing *Drosophila* nervous system. By sequencing a previously uncharacterised allele of Pbl the authors found that this allele represents isoform A of Pebble allowing them to generate novel tools to study the localisation and function of isoforms A and B separately. With this elegant approach the authors then characterised the regulation of the formation and dynamics of the cytokinetic furrow using neuroblasts as a model system. These stem cells divide asymmetrically in fate but also in size. The regulation of size asymmetry is not well understood in general. By using live cell imaging of mainly Myosin localisation in neuroblasts in different genetic conditions the authors propose differential functions of the isoforms in regulating the asymmetry of daughter cell size. The main proposition is that neuroblasts can sense and correct the position of the furrow to keep the much smaller neuroblast daughter cells at a constant size. The concepts are interesting, the tools and novel characterisation of Pbl exciting. This is a manuscript that should be published. However, there are a number of shortcomings that I would recommend to address. In particular, much of the concepts and working models proposed relies on quantifying fluorescent intensities and cell size. These need to be more robust to be fully convincing.

Figure 1

It is convincing that the "widths" of the Sqh "belt" (not the furrowing as depicted in the figure) as furrowing progresses is less in the mutant. It may well be that there are also lower Sqh levels at the forming GMC.

However, the quantifications of Sqh levels may be difficult as there are other cells abutting the relevant cortex of the neuroblast/GMC as shown in Fig 1 A. The quantification of cell size in vivo is always difficult. DIC as a criterion with other cells close by to measure GMC size seems not ideal.

1) Would it be possible to use a membrane marker (PH-GFP, or cytoplasmic GFP or similar) and high resolution confocal imaging to render the volume in cases where no confounding signal from other cells is present. This would be much more convincing. Same for figure 4L.

wt versus pbIMS. How is anaphase onset determined when looking at sqh? By the deformation of the cortex? Likewise, how was nuclear envelope reformation determined by the exclusion of sqh signal from the reforming nucleus? Using this as a criterion, it seems like the dynamics of nuclear reformation is affected? Delay in furrow formation seems plausible, but a delay of mitosis more unlikely although not impossible in the pbIMS mutants.

2) It would be good to determine if mitosis proceeds normally or with a delay in pbIMS neuroblasts looking at HistoneRFP or similar markers. It may have an impact on the comparison of the Sqh intensities and at the kinetics of furrowing that currently is claimed to occur normally based on the data shown in Figure 1F.

Points 1+2 would be good to strengthen to lend more convincing support for the idea of the correction mechanism (Fig 1K) that is proposed to not occur in the mutants. 3) Examples of wildtype cells with "severe basal position" and the frequency of their occurrence should be documented as they appear to be crucial to the study and the working model

The observation that pbIMS homozygous larvae have reduced negative geotaxis and life span is very interesting but no experiments link this directly to the size asymmetry

defects of neuroblast divisions as the entire animal is mutant for that allele. As presented to results are too suggestive of a causal link. Without further validation and data that are too much to ask for a revision this could perhaps better form part of another study. Why are pblMS homozygous mutants viable but the rescue construct (line 302) is not? Is the GFP perturbing function?

This obviously the authors choice, but why not swap figure 1 and 2? This could work nicely, then it is clear why the allele was used.

Figure 2

These are very significant and exciting findings! Very nice and convincing.

Figure 3

The main finding is that Pbl-A is at the cortex and goes to the cleavage furrow while Pbl-B is only at the poles and in the nuclei. This is very exciting and highly relevant. Yet the presentation of the data needs to be improved.

Some confusion with the description of the localisation. Line 268, I do not see Pbl-A in the nucleus? (video code: 400859_0_video_7116610_rlt58t and figure 3?). It is in the cytoplasm and the nucleus. Pbl-B is clearly sent to the nucleus.

4) It would be important to quantify the cortical localisations of the constructs in some way to provide a robust description of the differential localisation. Clarify what is meant by Pbl-A nuclear localisation? Frame 4:00 to 700 (video 400859_0_video_7116611_rlt58t) shows enrichment of Pbl-B at what looks like the furrow? Are there two rings made? On at the top of the "tubular shape" one at the bottom. The situation looks slightly more complex.

This is important as the localisation is then key to assess and assign differential function of the Pbl isoforms that is then tested next.

Figure 4

I am confused why when adding Pbl-A and B no localisation in the nucleus is detected. Figure 4D how was the position of the furrow determined? The quantifications in Figure 4G-K depend on that. The quantification of the Sqh levels here is not convincing. The movie of Pbl-B suggests that most of the "tubular shape" is absorbed by the neuroblast. Quantifying the area change of the GMC might be informative.

5) Line 316, I think it would be helpful to establish a reference for the timing of furrowing. If mitosis timing is unchanged (as suggested above) it would be good to use condensation/decondensation of chromatin as a reference point as the authors make a strong point that the rate of furrow ingression is not affected (line 316). Delays of the kinetics may well affect the interpretation. They even may have the data S3. I think this is critical as the study strongly suggests

Minor

Please add scale bars to all videos

Please add a minimum description of the live imaging method

Line 117 check subheading

Line 121 If not agreeing with suggestion above, please remind the reader of the nature of the pblMS allele already here and justify why this allele was chosen in the first place.

Line 126 please add time point of when "mid furrow ingression" is measured.

Line 130 nucleus??

Line 243 there could be some conclusion here

Line 276 please add transiently

Reviewer #2 (Remarks to the Author):

The mechanisms of asymmetric cell division have been well studied in model organisms including *C. elegans* (zygote) and *Drosophila* (neuroblasts). *Drosophila* neuroblasts provide a particularly appealing model based on the fact that the ganglion mother cells exhibit a dramatic size difference compared to the neuroblast. While the molecular machinery controlling neuroblast asymmetric divisions in the fly have been studied to a large extent, the cellular mechanisms that control the size difference during the cytokinetic cleavage have not been addressed in much detail. The manuscript by Royou and colleagues reports an important step forward towards resolving this problem. The authors demonstrate that one of the key players in cytokinesis, the Rho-GEF Pebble (Pbl), not only provides the well-known activation of Rho-dependent Myosin localisation and Formin-dependent F-actin polymerisation at the furrow, but also controls the maintenance and repositioning of the cytokinetic furrow before division. The authors show very elegantly that these two functions of Pbl are mediated by distinct protein isoforms, Pbl-A and Pbl-B, that originate from alternative splicing of the primary transcript.

By characterizing the mutant of pbl[M] allele, the authors show that neuroblasts lacking Pbl-B produced abnormally small ganglion mother cells, because the initial positioning of the cleavage furrow was maintained. They discover that a RacGAP50C-independent pathway mediated by Pbl-B is responsible for the control and refinement of cleavage furrow positioning. The authors further define functions of Pbl isoforms in controlling two subsets of Myosin accumulations – one at the cytokinetic furrow mediated by Pbl-A - and one at the entire cell cortex mediated by Pbl-B – both of which are required for robust production of ganglion mother cells with equally small cell sizes.

Furthermore, they show that in the absence of this quality control mechanism, adults exhibit neuronal malfunctions and decreased longevity. This study provides a major advance in our understanding of the molecular mechanisms of cytokinesis in asymmetrically dividing cells. The data show how distinct splice products of a gene serve distinct molecular mechanisms in a common biological process.

Some queries that the authors may wish to address:

(1) It would be interesting to add to the question whether the control and maintenance of cytokinetic furrow positioning only occurs in neuroblast asymmetric cell divisions. For example, the male sterile phenotype in pbl[M] mutants suggests that stem cell divisions in the germ line are affected. Is there evidence that Pbl-B serves similar functions in other stem cell divisions? Also, does the pbl[M] mutation exhibit phenotypes in other, non-stem-cell, cell types to suggest that Pbl-B plays a more general role in refining or maintenance of the cytokinetic furrow?

(2) A possibly related question. How do the authors interpret the fact that Pbl-B is able to rescue the absence of Pbl-A in amorphic pbl allelic combinations to a large extent? The authors discuss a role for polarity determinants in furrow positioning in neuroblast divisions. In case of the adult escapers that were mentioned, one would assume that Pbl-B is sufficient for cytokinesis in both stem cells and non-stem cells e.g. in the embryo or imaginal discs. Can the authors comment on this?

(3) Fig. 2E. The demonstration of the expression of Pbl-B and Pbl-A were done by using a transgenic construct and qRT-PCR analyses. With those tools at hand, the demonstration of the presence of the isoforms in addition to larval brains, e.g. testes, ovary, embryos, would have been beneficial to the scientific community.

Minor comments:

The quality of the data is excellent and the manuscript is very well written with little typos (I was wondering whether *Drosophila* was not consistently spelled with a capital D and italics).

In line 128 ff: I didn't understand the relationship between the cell cortex and the nucleus in the following sentence: "By the end of nuclear envelope reformation, myosin dissociated entirely from the cortex as illustrated by the depletion of myosin from the nucleus".

Reviewer #3 (Remarks to the Author):

This is a nice paper by Montembault et al. that examines how furrow position is regulated during the asymmetric divisions of *Drosophila* neuroblasts. The main result is the discovery of distinctly functioning splice isoforms of the RhoGEF Pbl. These isoforms have distinct localization that is consistent with one serving in its canonical role at the midzone/furrow and a predominantly cortical form that repositions the furrow towards the equator as it ingresses. Without the latter form the normally smaller sibling is too small. Overall, the study appears to be well-executed and the manuscript is nicely written. The contributions to the specific problem of asymmetric cell division (which already has broad appeal) and cytokinesis in general make me think that the work will have a high impact. I suggest publication with the minor changes described below.

- The authors conclude that small GMCs are eliminated in the pblms mutant but I couldn't find the evidence to support this claim. I suggest either adding the supporting evidence or removing the claim from the paper.

- The authors should provide quantifications to support the conclusions they make in the section comparing Pbl-A and Pbl-B localization (line 245).

- The authors note that "Importantly, GFP::Pbl-B dynamics were similar in the presence of endogenous Pbl" (line 276) but the data supporting this point is not shown and it is not clear what was similar and what was different. Furthermore, it wasn't clear to this reviewer which data was shown in Figure 3 (with or without endogenous Pbl).

- Does wild type Pbl have a localization pattern consistent with Pbl-A + Pbl-B (including the nuclear localization)?

- A slight modification to the title may be useful to readers - "Two RhoGEF isoforms with distinct localisation control furrow position during asymmetric cell division"

- If the authors already have the data in hand, it may be interesting to readers to comment on whether the myosin dynamics that occur outside of cytokinesis (PMID: 34779402) are differentially regulated by the Pbl isoforms

Reviewer #4 (Remarks to the Author):

This is a very interesting, well-written and elegant study that contributes major new insight on a number of important fronts of cell biology and developmental biology.

Through live imaging of asymmetrically dividing *Drosophila* neuroblasts, it reveals previously unappreciated complexity to the mechanisms of RhoA activation during cytokinesis. A second non-canonical Pebble RhoGEF isoform (Pbl-B) is uncovered, and the 2 isoforms (Pbl-A and Pbl-B) are shown to be differentially localized and regulated, with each being responsible for a different subset of Pbl-dependent behaviors. An elegant and compelling model is proposed for how the combined actions of Pbl-A and Pbl-B ensure that cytokinetic furrowing results in appropriate daughter cell size, while also providing mechanistic insight into the author's previous studies on the adaptability of the cell cortex in response to trailing chromatin in the cleavage plane. The data are of very high quality and the conclusions and title are mostly justified by the data (see below). The findings will be of great interest to many cell and developmental biologists working in the fields of cytokinesis, Rho GTPases, cytoskeletal remodelling, asymmetric division of stem cells and neuronal development. I recommend publication, with only a few minor points to address:

1) A clear strength of the manuscript is that most of the observations are well supported by quantification and statistical analysis. The only exception is Fig. 3 where the localization patterns of GFP::Pbl-A and GFP::Pbl-B are compared. Only a single time-lapse sequence per condition is shown, without quantification. Given the importance of the different localization patterns to the conclusions of the paper, it would seem appropriate and beneficial to provide some quantification here, both at the furrow and at the reforming nuclei. Also for GFP::Pbl-A the later nuclear localization should be shown for comparison. At present, it is unclear where the reforming nuclei are or whether they are simply missing from the optical section(s) shown.

2) There is a slight concern that pblMS may not be a null allele of the Pbl-B. Rather, a form truncated within the region encoded by exon 8 may be produced that might act as a partial dominant-negative, e.g. at the poles. "Pbl-A" is rather Pbl-A + Pbl-BΔC-term. The authors are careful to describe the experimental setup as "lacking a functional Pbl-B" but they could perhaps also more explicitly explain in the Discussion that the PblMS and "Pbl-A" pAcman genotypes may include some non-functional truncated Pbl-B.

3) Following on from point 2), have the authors tried targeting exon 8 by RNAi to selectively deplete the B isoform? Or to delete exon 8 in the pAcman-Pbl? These approaches would eliminate any concern regarding truncated Pbl-B, but I understand they are not trivial and are not essential for publication.

4) RacGAP50C RNAi is shown to convert Pbl-A localization to Pbl-B-like localization and furrowing proceeds in a Pbl-B-dependent manner. Does RacGAP50C RNAi also accelerate the nuclear localization of Pbl-A (to resemble Pbl-B)? This would seem to be a prediction of the model and could therefore further support the model. This would suggest that RacGAP50C-dependent targeting of Pbl-A to the midzone/ furrow delays its nuclear import.

5) "no homology to other protein domains was identified in the 457 amino-acid sequence encoded by exon 8". But how well conserved is this sequence, in other *Drosophila* species, and across phyla? It would be helpful to the reader to include a brief discussion of this.

Typos:

Ln250 "To do so we produced transgenic flies strains expressing GFP::Pbl-A or B under the endogenous pbl." Under the control of the endogenous pbl promoter?

In 485 "Pb-B" should be "Pbl-B"

REVIEWER COMMENTS

We are grateful for the constructive points that the four reviewers have raised. We hope they will find that the manuscript has now been strengthened in light of their input. We provide a point-by-point response below to the points that were raised.

Reviewer #1 (Remarks to the Author):

This study sheds light on the function of different isoforms of the evolutionarily conserved RhoGEF Pebble in flies in the context of the regulation of physical size asymmetry between daughter cells in stem cells of the developing *Drosophila* nervous system. By sequencing a previously uncharacterised allele of Pbl the authors found that this allele represents isoform A of Pebble allowing them to generate novel tools to study the localisation and function of isoforms A and B separately. With this elegant approach the authors then characterised the regulation of the formation and dynamics of the cytokinetic furrow using neuroblasts as a model system. These stem cells divide asymmetrically in fate but also in size. The regulation of size asymmetry is not well understood in general. By using live cell imaging of mainly Myosin localisation in neuroblasts in different genetic conditions the authors propose differential functions of the isoforms in regulating the asymmetry of daughter cell size. The main proposition is that neuroblasts can sense and correct the position of the furrow to keep the much smaller neuroblast daughter cells at a constant size. The concepts are interesting, the tools and novel characterisation of Pbl exciting. This is a manuscript that should be published. However, there are a number of shortcomings that I would recommend to address. In particular, much of the concepts and working models proposed relies on quantifying fluorescent intensities and cell size. These need to be more robust to be fully convincing.

We thank the reviewer for this positive evaluation.

Figure 1

It is convincing that the "widths" of the Sqh "belt" (not the furrowing as depicted in the figure) as furrowing progresses is less in the mutant. It may well be that there are also lower Sqh levels at the forming GMC.

However, the quantifications of Sqh levels may be difficult as there are other cells abutting the relevant cortex of the neuroblast/GMC as shown in Fig 1 A. The quantification of cell size in vivo is always difficult. DIC as a criterion with other cells close by to measure GMC size seems not ideal.

1) Would it be possible to use a membrane marker (PH-GFP, or cytoplasmic GFP or similar) and high resolution confocal imaging to render the volume in cases where no confounding signal from other cells is present. This would be much more convincing. Same for figure 4L.

The reviewer raises concerns about the quantification of Sqh::GFP fluorescence intensities and the use of DIC for cell size measurements due to cells abutting the relevant cortex. To

allay these concerns, the reviewer suggested repeating the GMC size measurements as shown in fig. 1h, 4l and 5j using a cytoplasmic or membrane-associated GFP marker “in cases where no confounding signal from other cells is present”. Two misunderstandings could have led the reviewer to question the robustness of our GFP intensity measurements and cell size quantification: (1) our choice of data presentation in fig. 1a, 4c and 5b, where maximum projections of Sqh::GFP signal were shown instead of a sagittal view, (2) the fact that a detailed method for fluorescence intensity measurements and cell size quantification was not provided in the original submission. Here, we provide an explanation that we hope will resolve these misunderstandings and allay their concern:

(1) Maximum projections are shown in fig. 1a, 4c and 5b (and the associated videos) because, depending on the cell, the axis of division does not remain exactly parallel to the XY plane throughout cytokinesis. In order to illustrate myosin dynamics the most accurately in both the nascent neuroblast and the GMC without losing too much information for the reader, we chose to present maximum projections of the relevant z planes in the figures (and video). As a result, the GMC cortex in the WT cell shown in fig. 1a, for instance, is not as distinct as it would have been had we opted to display a single z image (see fig. 1b for an example of the sagittal view of a single z section). This also applies to myosin exclusion from the nucleus at the end of cytokinesis not being as visible in the *pbl^{MS}* mutant as it is in the WT in fig. 1a.

(2) However, all quantification of GFP signal and GMC size measurements were carried out on a single z plane of the most sagittal view of the relevant nascent cell. We have included this point in the Methods section (lines 636 and 661-5). We measured the area of the GMC using the Sqh::GFP (myosin) signal (see light orange coloured ROI in the figure below), since the Sqh::GFP signal is mostly cytoplasmic at this time point. Thus, the Sqh::GFP serves as the cytoplasmic marker that the reviewer requests. We also verified that the ROI delineating the GMC area on the Sqh::GFP signal corresponded to the GMC in the DIC channel, which adds another level of accuracy “in case of confounding Sqh::GFP signal from other cells is present”. We are therefore confident that this method of measurement accurately reflects the area of the GMC. As the reviewer will see in the image below, the difference in GMC cell size between WT and *pbl^{MS}* is clear. Since the GMC forms a regular sphere at the end of furrow ingression (see images below), the use of surface area to reflect cell size seems appropriate. Moreover, previous studies have used 2-dimensional images to measure cell size (see PMIDs 34706235 and 29123099 for two recent examples of GMC size measurements). Measuring the 3-dimensional volume would be much more technically challenging. Finally, the fact that we find similar GMC size distribution between WT cells and cells expressing the full length *pbl* genomic construct (Pbl-A+B) as well as *pbl^{MS}* cells and cells expressing Pbl-A, a mimic of the *pbl^{MS}* allele, we feel that the data underscores the robustness of our method for GMC size measurements (see scatter dot plots below, bars indicate median \pm interquartile range).

For clarification, we have amended Fig. 1g by outlining the GMC with a lightly-coloured ROI on the Sqh::GFP channel and shown the corresponding DIC channel. In addition, we added the following sentence in the methods section: “GMC area plotted in fig. 1h, 4m and 5j were determined as follows: a ROI delineating the GMC on a single z image of the most sagittal view using the Sqh::GFP signal was drawn and the number of pixels within the ROI determined. The accuracy of the ROI was confirmed using the DIC channel.” (lines 661-5).

wt versus pb1MS. How is anaphase onset determined when looking at sqh? By the deformation of the cortex? Likewise, how was nuclear envelope reformation determined by the exclusion of sqh signal from the reforming nucleus? Using this as a criterion, it seems like the dynamics of nuclear reformation is affected? Delay in furrow formation seems plausible, but a delay of mitosis more unlikely although not impossible in the pb1MS mutants.

2) It would be good to determine if mitosis proceeds normally or with a delay in pb1MS neuroblasts looking at HistoneRFP or similar markers. It may have an impact on the comparison of the Sqh intensities and at the kinetics of furrowing that currently is claimed to occur normally based on the data shown in Fig. 1F.

In all our figures, the reference time (0:00) corresponds to anaphase onset (defined as the initiation of sister chromatid separation) except for fig. 5a where time 0:00 corresponds to the onset of furrow ingression (defined as the onset of inward membrane curvature). Anaphase onset is determined using the DIC channel since chromosomes are readily detected using DIC illumination (an example of a cell expressing H2A.Z::mRFP and Sqh::GFP (Myosin) and the corresponding DIC image is provided below to illustrate this point to the reviewer). We have included the following sentence in the methods section for clarification: “DIC images and the relevant fluorescence images were acquired near-simultaneously for all time points. Since chromosomes are readily detected by DIC illumination, we used anaphase onset, defined as the initiation of sister chromatid separation, as the reference (time, 0:00) in all our figures except fig. 5a where time, 0:00 refers to the onset of furrow ingression” (lines 623-27).

We have substantially changed the graphs in fig. 1f, 4k and 5h to show the % of ring constriction over time instead of the furrow width. We considered 0% ingression at anaphase onset and 100% ingression when the ring reached the midbody stage (3 to 5 pixels diameter). For transparency, we displayed the measurements of each individual cell (dashed lines) and the average (solid lines).

Points 1+2 would be good to strengthen to lend more convincing support for the idea of the correction mechanism (Fig 1K) that is proposed to not occur in the mutants. 3) Examples of wildtype cells with “severe basal position” and the frequency of their occurrence should be documented as they appear to be crucial to the study and the working model

As requested by the reviewer, we have provided examples of normal and severe basal positions of the furrow at the onset of ingression for each genotype and the resulting GMC size at the end of furrow ingression (new fig. 1k). The frequency of their occurrence is documented in the original and revised submission in fig. 1i and 1j, where each dot represents a cell. If we arbitrary consider that the furrow has a severe basal position at onset of ingression when the distance between furrow and basal pole is less than 36 pixels then, according to fig. 1j, the frequencies of WT and *pbl^{MS}* cells with furrow severe basal position are 42% and 33% respectively.

The observation that *pblMS* homozygous larvae have reduced negative geotaxis and life span is very interesting but no experiments link this directly to the size asymmetry defects of neuroblast divisions as the entire animal is mutant for that allele. As presented to results are too suggestive of a causal link. Without further validation and data that are too much to ask for a revision this could perhaps better form part of another study.

As requested, we have removed the data on geotaxis and survival from fig. 1l and m in the revised manuscript submission.

Why are *pblMS* homozygous mutants viable but the rescue construct (line 302) is not? Is the GFP perturbing function?

In fig. 4, the construct that mimics *pbl^{MS}* is called Pbl-A and this construct is viable. There is no GFP associated with these constructs.

This obviously the authors choice, but why not swap fig. 1 and 2? This could work nicely, then it is clear why the allele was used.

The reviewer's point is well-taken. However, we would prefer to keep the original ordering of the figures since we prefer to present a thorough analysis of the *pbl^{MS}* phenotype (fig. 1), before presenting the identification of the point mutation responsible for the phenotype (fig. 2). An additional paragraph in the introduction described *pbl^{MS}* as an hypomorph mutation that we have used in two previous studies (PMID: 28835609 and 23185030).

Figure 2

These are very significant and exciting findings! Very nice and convincing.

We thank the reviewer for this encouraging comment.

Figure 3

The main finding is that Pbl-A is at the cortex and goes to the cleavage furrow while Pbl-B is only at the poles and in the nuclei. This is very exciting and highly relevant. Yet the presentation of the data needs to be improved.

Some confusion with the description of the localisation. Line 268, I do not see Pbl-A in the nucleus? (video code: 400859_0_video_7116610_rlt58t and figure 3?). It is in the cytoplasm and the nucleus. Pbl-B is clearly sent to the nucleus.

4) It would be important to quantify the cortical localisations of the constructs in some way to provide a robust description of the differential localisation. Clarify what is meant by Pbl-A nuclear localisation?

As requested by the reviewer, we have now improved fig. 3, by providing the quantification of GFP::Pbl-A and B mean intensity at the midzone, furrow and polar membrane at the onset of furrow ingression (new fig. 3b, c, d and e respectively) and at the midbody, nucleus and polar membrane at the end of furrow ingression (new fig. 3f, g, h and i respectively). A detailed method of how the mean gray value was quantified for each region is provided in the Methods section (lines 645-50).

Frame 4:00 to 700 (video 400859_0_video_7116611_rlt58t) shows enrichment of Pbl-B at what looks like the furrow? Are there two rings made? On at the top of the "tubular shape" one at the bottom. The situation looks slightly more complex.

The reviewer is probably referring to fig. 4c in this comment. Fig. 4c shows myosin dynamics in *pbl³/pbl²* mutant cells expressing the indicated Pbl genomic constructs (described in fig. 4a) and not GFP::Pbl-A or B or both. In order to make this point clearer to readers, we have now labelled the legend *Pbl-A+B*, *Pbl-A* and *Pbl-B* in italics in order to underscore the fact that we are referring to a genotype and not to a GFP fusion in the revised fig. 4 and in the text.

Figure 4

I am confused why when adding Pbl-A and B no localisation in the nucleus is detected.

The marker used in fig. 4 is Sqh::GFP (referred to as myosin in the figure), the myosin marker, which is excluded from the nucleus upon nuclear envelop reformation. See comment above.

Figure 4D how was the position of the furrow determined?

In the original submission, we had provided the following description for determining furrow length: "The length of the furrow was determined by the divergent angle from the long axis of the cell being $<30^\circ$ " (lines 653-4). In fig. 4g we provided an additional scheme showing a Pbl-B expressing cell with an elongated furrow to show what we considered the furrow (red line) and the polar cortex (blue line).

The quantifications in Figure 4G-K depend on that. The quantification of the Sqh levels here is not convincing. The movie of Pbl-B suggests that most of the "tubular shape" is absorbed by the neuroblast. Quantifying the area change of the GMC might be informative.

The zone on the "tubular shape" where the membrane eventually pinches varies from cell to cell. The "tubular shape" "is "absorbed" either by the neuroblast or the GMC. It is very dynamic, and therefore variable. Given this variability, it is unlikely that measuring the area change of the GMC will provide any additional insight. The two daughter cells eventually round back up at the end of ingression, without altering the overall GMC size (as illustrated in fig. 4m).

5) Line 316, I think it would be helpful to establish a reference for the timing of furrowing. If mitosis timing is unchanged (as suggested above) it would be good to use condensation/decondensation of chromatin as a reference point as the authors make a strong point that the rate of furrow ingression is not affected (line 316). Delays of the kinetics may well affect the interpretation. They even may have the data S3. I think this is critical as the study strongly suggests

We use anaphase onset, the initiation of sister chromatid separation, as time 0:00 in all our experiments except fig. 5a in the revised manuscript submission. The chromosomes are readily detected in the DIC channel (see comment and figure above), thus providing a robust reference point for subsequent temporal measurements.

Minor

Please add scale bars to all videos

Thank you for this comment. We have added the scale bar to the videos.

Please add a minimum description of the live imaging method

We have added a more detailed method for our live imaging of dissociated neuroblasts (line 608-16).

Line 117 check subheading

We do not know how the reviewer would like us to amend this subheading. However, we would be happy to make the subheading clearer for readers if you could provide us with some guidance.

Line 121 If not agreeing with suggestion above, please remind the reader of the nature of the *pbl^{MS}* allele already here and justify why this allele was chosen in the first place.

We have added a summary of our findings in the last paragraph of the introduction where the *pbl^{MS}* allele is introduced.

Line 126 please add time point of when “mid furrow ingression” is measured.

The onset of myosin polar cortex enrichment varies between cells, therefore we amended the sentence as follows: " Shortly after the onset of furrow ingression (defined as onset of membrane inward curvature), a pool of myosin underwent outward flow from the ring and enriched the entire cortex of both nascent daughter cells".

Line 130 nucleus??

This sentence has been changed as follows: “Myosin dissociated entirely from the cortex concomitantly with nuclear envelope reassembly (as illustrated by the depletion of myosin from the cortex and its exclusion from the nascent nucleus)”.

Line 243 there could be some conclusion here

At the end of the paragraph, we added the following sentence: “Since the folding of the three BRCTs are predicted to be preserved in Pbl-B, we anticipated that Pbl-A and B will exhibit similar dynamics during neuroblast division” (lines 261-63). This sentence provides a transition for the next paragraph, which documents the localisation of GFP::Pbl-A and B during cytokinesis.

Line 276 please add transiently

We do not understand where the word "transiently" could fit in line 276 of the original submission. Could the reviewer provide some guidance for us?

Reviewer #2 (Remarks to the Author):

The mechanisms of asymmetric cell division have been well studied in model organisms including *C. elegans* (zygote) and *Drosophila* (neuroblasts). *Drosophila* neuroblasts provide a particularly appealing model based on the fact that the ganglion mother cells exhibit a

dramatic size difference compared to the neuroblast. While the molecular machinery controlling neuroblast asymmetric divisions in the fly have been studied to a large extent, the cellular mechanisms that control the size difference during the cytokinetic cleavage have not been addressed in much detail. The manuscript by Royou and colleagues reports an important step forward towards resolving this problem. The authors demonstrate that one of the key players in cytokinesis, the Rho-GEF Pebble (Pbl), not only provides the well-known activation of Rho-dependent Myosin localisation and Formin-dependent F-actin polymerisation at the furrow, but also controls the maintenance and repositioning of the cytokinetic furrow before division. The authors show very elegantly that these two functions of Pbl are mediated by distinct protein isoforms, Pbl-A and Pbl-B, that originate from alternative splicing of the primary transcript.

By characterizing the mutant of pbl[M] allele, the authors show that neuroblasts lacking Pbl-B produced abnormally small ganglion mother cells, because the initial positioning of the cleavage furrow was maintained. They discover that a RacGAP50C-independent pathway mediated by Pbl-B is responsible for the control and refinement of cleavage furrow positioning. The authors further define functions of Pbl isoforms in controlling two subsets of Myosin accumulations – one at the cytokinetic furrow mediated by Pbl-A - and one at the entire cell cortex mediated by Pbl-B – both of which are required for robust production of ganglion mother cells with equally small cell sizes.

Furthermore, they show that in the absence of this quality control mechanism, adults exhibit neuronal malfunctions and decreased longevity. This study provides a major advance in our understanding of the molecular mechanisms of cytokinesis in asymmetrically dividing cells. The data show how distinct splice products of a gene serve distinct molecular mechanisms in a common biological process.

We thank the reviewer for these positive comments.

Some queries that the authors may wish to address:

(1) It would be interesting to add to the question whether the control and maintenance of cytokinetic furrow positioning only occurs in neuroblast asymmetric cell divisions. For example, the male sterile phenotype in pbl[M] mutants suggests that stem cell divisions in the germ line are affected. Is there evidence that Pbl-B serves similar functions in other stem cell divisions? Also, does the pbl[M] mutation exhibit phenotypes in other, non-stem-cell, cell types to suggest that Pbl-B plays a more general role in refining or maintenance of the cytokinetic furrow?

The reviewer raises an excellent idea that we are currently investigating in the lab. The study of the contribution of each isoform during cytokinesis in other tissues is highly relevant. We hope to provide a thorough analysis of cytokinesis in spermatocytes, epithelium and other cell types in the future, but we hope the reviewer will agree that developing these different tissue models adequately is beyond the scope of the current paper.

(2) A possibly related question. How do the authors interpret the fact that Pbl-B is able to rescue the absence of Pbl-A in amorphic pbl allelic combinations to a large extent? The

authors discuss a role for polarity determinants in furrow positioning in neuroblast divisions. In case of the adult escapers that were mentioned, one would assume that Pbl-B is sufficient for cytokinesis in both stem cells and non-stem cells e.g. in the embryo or imaginal discs. Can the authors comment on this?

We are currently assessing the contribution of each isoform to cell divisions exhibited by different cell types in order to hone in on how different Pbl isoforms contribute to specific tissue features. Since embryos lacking either Pbl-A or B can develop into adults, albeit with varying success (fig. 4b), we propose that the two isoforms play redundant functions, including promoting cytokinesis regardless of cell type. However, we have preliminary data suggesting that epithelial cells lacking Pbl-A have an increased incidence of cytokinesis failure. Conversely, spermatocytes lacking Pbl-B fail cytokinesis, leading to male sterility. We are now focused on understanding the mechanisms underlying these cytokinesis defects.

(3) Fig. 2E. The demonstration of the expression of Pbl-B and Pbl-A were done by using a transgenic construct and qRT-PCR analyses. With those tools at hand, the demonstration of the presence of the isoforms in addition to larval brains, e.g. testes, ovary, embryos, would have been beneficial to the scientific community.

In the revised manuscript, we have added the qRT-PCR and RNAseq analysis for the adult testes (fig. 2c). Additionally, we have provided a table presenting the number of reads of exon junction usage and the corresponding percentage from the RNAseq analysis for embryos, whole adult, larval CNS, adult ovary and testes (Table S1). We have also provided the raw data RNAseq analysis that includes reads from additional tissues.

Minor comments:

The quality of the data is excellent and the manuscript is very well written with little typos (I was wondering whether *Drosophila* was not consistently spelled with a capital D and italics).

We have endeavoured to correct the typos and make the use of "*Drosophila*" consistent.

In line 128 ff: I didn't understand the relationship between the cell cortex and the nucleus in the following sentence: "By the end of nuclear envelope reformation, myosin dissociated entirely from the cortex as illustrated by the depletion of myosin from the nucleus".

This sentence has been changed in the revised manuscript as follows: "Myosin dissociated entirely from the cortex concomitantly with nuclear envelope reassembly (as illustrated by the depletion of myosin from the cortex and its exclusion from the nascent nucleus)" (lines 144-6).

Reviewer #3 (Remarks to the Author):

This is a nice paper by Montembault et al. that examines how furrow position is regulated during the asymmetric divisions of *Drosophila* neuroblasts. The main result is the discovery of distinctly functioning splice isoforms of the RhoGEF Pbl. These isoforms have distinct

localization that is consistent with one serving in its canonical role at the midzone/furrow and a predominantly cortical form that repositions the furrow towards the equator as it ingresses. Without the latter form the normally smaller sibling is too small. Overall, the study appears to be well-executed and the manuscript is nicely written. The contributions to the specific problem of asymmetric cell division (which already has broad appeal) and cytokinesis in general make me think that the work will have a high impact. I suggest publication with the minor changes described below.

We thank the reviewer for their encouraging evaluation of the manuscript.

- The authors conclude that small GMCs are eliminated in the *pblms* mutant but I couldn't find the evidence to support this claim. I suggest either adding the supporting evidence or removing the claim from the paper.

This data has been removed from the resubmission.

- The authors should provide quantifications to support the conclusions they make in the section comparing Pbl-A and Pbl-B localization (line 245).

As requested, we have now provided the quantification of GFP::Pbl signal at the midzone, furrow and polar membrane (fig. 3b-e) at onset of furrow ingression. We also provide fluorescence quantification of GFP::Pbl at the midbody, nucleus and polar membrane at the end of furrow ingression (fig. 3f-i). See comment above.

- The authors note that "Importantly, GFP::Pbl-B dynamics were similar in the presence of endogenous Pbl" (line 276) but the data supporting this point is not shown and it is not clear what was similar and what was different. Furthermore, it wasn't clear to this reviewer which data was shown in Figure 3 (with or without endogenous Pbl).

The legend of fig. 3a has been changed to indicate that the cell is without the endogenous Pbl (transheterozygote null alleles *pbl²/pbl³*). In addition, the scatter dot plots provided in fig. 3c-i show the quantification of GFP::Pbl-A and B levels in both WT and *pbl²/pbl³* backgrounds (light and dark dots respectively).

- Does wild type Pbl have a localization pattern consistent with Pbl-A + Pbl-B (including the nuclear localization)?

As expected from the western in fig. 2e, the localisation of GFP::Pbl genomic is consistent with Pbl-A + Pbl-B (see images below for an example of neuroblasts expressing GFP::Pbl genomic vs GFP::Pbl-A vs GFP::Pbl-B). However, we did not include this data in the revised manuscript, as we do not feel that it provides additional insight to the work.

- A slight modification to the title may be useful to readers - "Two RhoGEF isoforms with distinct localisation control furrow position during asymmetric cell division"

We thank the reviewer for this title suggestion and have changed it accordingly.

- If the authors already have the data in hand, it may be interesting to readers to comment on whether the myosin dynamics that occur outside of cytokinesis (PMID: 34779402) are differentially regulated by the Pbl isoforms

We completely agree with the reviewer that these questions are very interesting and we plan on pursuing them in future studies. For this paper we would like to remain focused on cytokinesis in order to provide a clear and succinct message for readers.

Reviewer #4 (Remarks to the Author):

This is a very interesting, well-written and elegant study that contributes major new insight on a number of important fronts of cell biology and developmental biology. Through live imaging of asymmetrically dividing *Drosophila* neuroblasts, it reveals previously unappreciated complexity to the mechanisms of RhoA activation during cytokinesis. A second non-canonical Pebble RhoGEF isoform (Pbl-B) is uncovered, and the 2 isoforms (Pbl-A and Pbl-B) are shown to be differentially localized and regulated, with each being responsible for a different subset of Pbl-dependent behaviors. An elegant and compelling model is proposed for how the combined actions of Pbl-A and Pbl-B ensure that cytokinetic furrowing results in appropriate daughter cell size, while also providing mechanistic insight into the author's previous studies on the adaptability of the cell cortex in response to trailing chromatin in the cleavage plane. The data are of very high quality and the conclusions and title are mostly justified by the data (see below). The findings will be of great interest to many cell and developmental biologists working in the fields of cytokinesis, Rho GTPases, cytoskeletal remodelling, asymmetric division of stem cells and neuronal development. I recommend publication, with only a few minor points to address:

We thank the reviewer for this positive evaluation of the work.

1) A clear strength of the manuscript is that most of the observations are well supported by quantification and statistical analysis. The only exception is Fig. 3 where the localization patterns of GFP::Pbl-A and GFP::Pbl-B are compared. Only a single time-lapse sequence per condition is shown, without quantification. Given the importance of the different localization patterns to the conclusions of the paper, it would seem appropriate and beneficial to provide some quantification here, both at the furrow and at the reforming nuclei. Also for

GFP::Pbl-A the later nuclear localization should be shown for comparison. At present, it is unclear where the reforming nuclei are or whether they are simply missing from the optical section(s) shown.

As requested, the quantification of GFP::Pbl signal has been added to fig. 3. Specifically, we quantified GFP::Pbl signal at the midzone, furrow and polar membrane (fig. 3b-e) at the onset of furrow ingression and the midbody, nucleus and polar membrane at the end of furrow ingression (fig. 3f-i).

2) There is a slight concern that *pbl^{MS}* may not be a null allele of the Pbl-B. Rather, a form truncated within the region encoded by exon 8 may be produced that might act as a partial dominant-negative, e.g. at the poles. “Pbl-A” is rather Pbl-A + Pbl-BΔC-term. The authors are careful to describe the experimental setup as “lacking a functional Pbl-B” but they could perhaps also more explicitly explain in the Discussion that the *pbl^{MS}* and “Pbl-A” pAcman genotypes may include some non-functional truncated Pbl-B.

We agree with the reviewer. We added in the discussion: “The premature stop codon in exon 8 prevents the expression of a functional Pbl-B in the *pbl^{MS}* mutant. Since mRNA with premature stop codons are often rapidly degraded by the nonsense-mediated mRNA decay (NMD) pathway, the *pbl^{MS}* phenotypes observed during neuroblast divisions are likely the result of a loss of Pbl-B function rather than the dominant effect of truncated Pbl-B expression. However, we can not formally exclude this possibility.”

3) Following on from point 2), have the authors tried targeting exon 8 by RNAi to selectively deplete the B isoform? Or to delete exon 8 in the pAcman-Pbl? These approaches would eliminate any concern regarding truncated Pbl-B, but I understand they are not trivial and are not essential for publication.

We have expressed GFP::Pbl-A in the *pbl³/pbl²* background and we found that it recapitulated the *pbl^{MS}* phenotype, including no myosin polar cortex enrichment and the production of a subset of abnormally small GMC. However, we have not included these results in the manuscript because they are currently preliminary.

4) RacGAP50C RNAi is shown to convert Pbl-A localization to Pbl-B-like localization and furrowing proceeds in a Pbl-B-dependent manner. Does RacGAP50C RNAi also accelerate the nuclear localization of Pbl-A (to resemble Pbl-B)? This would seem to be a prediction of the model and could therefore further support the model. This would suggest that RacGAP50C-dependent targeting of Pbl-A to the midzone/ furrow delays its nuclear import.

No, we did not find that RacGAP RNAi accelerated the nuclear localisation of Pbl-A. We revised fig. 5a to illustrate this point by replacing the last image by an image at a later time point in the same cell showing the faint localisation of GFP::Pbl-A in the nucleus. In addition, we included time-lapse images of GFP::Pbl-B localisation upon RacGAP RNAi at equivalent time points for comparison. A strong GFP::Pbl-B signal is rapidly detected in the nucleus after initiation of furrow ingression.

5) “no homology to other protein domains was identified in the 457 amino-acid sequence

encoded by exon 8". But how well conserved is this sequence, in other *Drosophila* species, and across phyla? It would be helpful to the reader to include a brief discussion of this.

We added three supplementary figures (fig. S2-4) with this information and the following paragraph in the results section: "To examine the conservation of exon 8, we computed a multiple sequence alignment of 948 Pbl protein orthologs available in the OrhoDB database⁴². As expected, this alignment indicated high conservation of the whole Pbl sequence amongst Eukaryotes with the exception of the region spanning exon 8. The sequence including exon 8 is found in a subset of 233 sequences belonging almost exclusively to the insecta class (fig. S2). Furthermore, a short region within the exon 8 sequence is shared between *Drosophilidae* and mosquitoes (fig. S3). Finally, amongst *Drosophilidae*, exon 8 shows higher diversity than the rest of the Pbl sequence and is present in 36 out of 38 *Drosophilidae* species (fig. S4)." (lines 238-47).

Typos:

Ln250 "To do so we produced transgenic flies strains expressing GFP::*Pbl*-A or B under the endogenous *pbl*." Under the control of the endogenous *pbl* promoter?

Thank you. This has been corrected.

In 485 "*Pb*-B" should be "*Pbl*-B"

Thank you. This has been corrected.

Reviewer #1 (Remarks to the Author):

The authors have done a nice job in revising the manuscript and rebutting the concerns. The manuscript is now suitable for publication in my opinion. The proposal that these asymmetrically dividing cells have a mechanism to sense the size of the daughter cells via a regulator of Rho kinase activity is very interesting. The claims are backed up by well presented data and carefully executed experiments.

Reviewer #2 (Remarks to the Author):

The authors have addressed all my comments to their first submission. I can agree that some points raised might go beyond the scope of this manuscript and I am looking forward as to how this subject is going to develop in the future.

Reviewer #3 (Remarks to the Author):

The authors adequately addressed the comments on the initial submission and the revision is suitable for publication.

Reviewer #4 (Remarks to the Author):

All my points have been more than satisfactorily addressed and clearly highlighted in the revised (and improved) manuscript which, in my opinion, is ready for publication. In particular, the quantification of the GFP::Pbl-A/B patterns of localization presented in Fig. 3 is clear and compelling and strengthens the conclusions. The new text and supplementary figures regarding the conservation of the sequence in exon 8 is an interesting and appreciated addition. I congratulate the authors on a high-quality paper that makes a major contribution to our understanding of the regulation of cytokinesis and cell-size asymmetry in an important model system. It also provides a very interesting example of how evolution has selected partially redundant splice variants to fine-tune such a dynamic system and render it robust. This is probably much more common than currently appreciated.